# Effect of UV-A on endophyte colonisation of *Arabidopsis thaliana*

**Aleksandra Giza[1,2], Paweł Hermanowicz[1], Rafał Ważny[1], Agnieszka Domka[1,3], Piotr Rozpądek[1], Justyna Łabuz[1]***

1 Malopolska Centre of Biotechnology, Jagiellonian University, Kraków, Poland, 2 Doctoral School of Exact and Natural Sciences, Jagiellonian University, Kraków, Poland, 3 W. Szafer Institute of Botany, Polish Academy of Sciences, Kraków, Poland.

email address:
a.giza@doctoral.uj.edu.pl
pawel.hermanowicz@uj.edu.pl
rafal.wazny@uj.edu.pl
a.domka@botany.pl
piotr.rozpadek@uj.edu.pl
* justyna.sojka@uj.edu.pl (JL)

## Abstract

UV-A, an important part of sunlight radiation, is typically absent in experiments on plant-endophyte interactions. We examined the impact of UV-A in the 350–400 nm range (UV-A1 waveband) on the plant interactions with fungal endophytes belonging to different taxonomic groups: *Paraphoma chrysanthemicola*, *Phomopsis columnaris*, *Diaporthe eres*, *Mucor* sp., and yeast *Sporobolomyces ruberrimus*. Physiologically relevant levels of UV-A did not substantially affect the colonisation of shoots and roots by endophytes. UV-A upregulated the expression of genes involved in the establishment of symbiosis. Specifically, the expression of *PDF1.2* was affected by *P. chrysanthemicola* and *S. ruberrimus* only under UV-A conditions. Additionally, UV-A exposure upregulated the mRNA levels of *ICS1* and *PAL1*, genes important for plant responses to stress factors. Inoculation with *P. chrysanthemicola* and *S. ruberrimus* led to increased expression of the *ICS1* gene. We did not observe significant interactions between the effects of UV-A and the presence of endophytes on other examined plant traits, including plant fresh weight, root system architecture, and expression of plant photoreceptor genes. For these physiological parameters, the effects of the presence of endophytes did not depend on UV-A supplementation. Our findings indicate that while UV-A does not substantially influence plant colonisation by the endophytes, it does trigger the upregulation of plant defence genes and affects the shoot growth of *Arabidopsis*.

**Data availability statement:** Data analysed during this study are included in this published article and its supplementary information files. The raw data for figures 2 - 6 are available through FigShare (https://figshare.com/articles/dataset/Effect_of_UV-A_on_Endophyte_Colonisation_of_Arabidopsis_thaliana_raw_data/27284010). The nucleotide sequences of microorganisms are available at NCBI database under accession numbers OR077971 (P. chrysanthemicola), OR077985 (P. columnaris), OR078011 (D. eres), OR078038 (Mucor sp.) and ON331994 (S. ruberrimus).

**Funding:** This research was funded by the Priority Research Area BioS under the program Excellence Initiative – Research University at the Jagiellonian University in Krakow, project "The role of ultraviolet radiation in the colonisation of plants by endophytes", number B.1.11.2020.74.

## Introduction

Light is an important factor directly controlling the growth and development of plants, which also modulates responses to biotic and abiotic stresses [1]. The shortest wavelengths in sunlight, UV-B (280–320 nm) and UV-A radiation (320–400 nm), have detrimental effects on plant fitness due to interaction with DNA and proteins [2]. However, at lower irradiances, UV is an important environmental cue, affecting gene expression, physiology, metabolite accumulation, and morphology [3]. Although UV-A predominates over UV-B in the solar spectrum, its impact on plant morphology and growth is less straightforward [4]. The response to UV-A appears to depend on the species, the dose, and other environmental factors. UV-A has a stimulatory effect on *Arabidopsis thaliana* rosette size, in contrast to the inhibitory effect caused by UV-B exposure [5]. In *Arabidopsis,* the impact of UV-A strongly depends on the ecotype, resulting in either reduced or enhanced plant biomass production [6]. In tomatoes, UV-A increases leaf area and promotes growth [7]. In lettuce, leaf area decreases, but biomass increases upon UV-A treatment [8]. In cucumber, UV-A supplementation results in compact and sturdy plant phenotype [9]. UV-A also induces the accumulation of flavonoids and their derivatives in a dose-dependent manner [10]. A UV-A-associated increase in total phenolic content has been reported for green-leaf pakchoi [11], red-leaf lettuce [12], and broccoli sprouts [13]. Flavonoids serve as sunscreens protecting plants from damage [14], but also act as chemo-attractants for soil bacteria [15].

UV radiation is perceived by UV and blue light photoreceptors. Phototropins (phot1 and phot2) [16], and cryptochromes (cry1 and cry2) [17] absorb UV-A due to their chromophores, flavin mononucleotide (FMN) or flavin adenine dinucleotide (FAD) and 5, 10-methyltetrahydrofolate (pterin) [18,19], respectively. These receptors absorb mainly the UV-A1 waveband (350–400 nm) and blue light. The UVR8 receptor (UV RESISTANCE LOCUS 8) perceives UV-B and partially shorter-wavelength UV-A (i.e., UV-A2, 315–350 nm) thanks to its tryptophane residues [20–23]. Cryptochromes and UVR8 share the signalling pathway [21], being the major regulators of gene expression upon short wavelength irradiation [22]. Cryptochromes are also the key players in biotic and abiotic stress-related phenolic compound accumulation in plants exposed to blue light and UV-A. UVR8 is responsible for accumulating flavonoids in low UV-A conditions [24]. Phototropins are receptor kinases, which operate through protein phosphorylation to control rapid movements and growth responses [25]. Not much is known about their signalling triggered by UV radiation, apart from UV-B-induced chloroplast movements [26] and phototropism [27,28]. Phot1 partially regulates the accumulation of phenolic compounds, but only when blue light is absent [24].

UV-A is an essential component of the solar spectrum, but its role in plant physiology is much less studied than UV-B [4]. Few studies investigate the effect of UV-A on photosynthesis [29], reporting a positive impact [30,31]. Due to safety issues, UV-emitting lamps are rarely used in growth chambers, while the amount of ultraviolet present in greenhouses is reduced by the absorption by windowpanes and screens [2,32]. Such conditions substantially differ from those encountered in the natural environment, affecting plant physiology, morphology, secondary metabolism,

and interactions with other organisms. Not much is known about the role of UV in plant-microorganism interactions, their effects on UV perception, and growth of plants [33–36].

In recent years the interaction between plants and microorganisms has been extensively studied due to the high potential of PGPM (**P**lant **G**rowth **P**romoting **M**icroorganisms) in agriculture. These interactions rely on the mutual exchange of resources between the interacting organisms. Plants are a source of reduced carbon for their microbiota which becomes an additional carbon sink within the holobiont. Microorganisms provide their host with necessary nutrients, such as N, P, and K. The establishment of the interaction between plants and their microbiota requires significant mutual metabolic adaptations that balance the needs of the interacting organisms. One of them is the adaptation of plant energy production and distribution. The exchange of photo-assimilates between the plant and symbiotic microorganisms determines the relationship between the host and its microbiota and may cause alterations in microorganism lifestyle leading to dysbiosis. Thus, factors affecting light conversion into chemical energy may impact the relationship between plants and their microbiota. Another important interaction-related adaptation is the control of microorganism progression inside plant tissues by a specific immune response. Selected defence mechanisms controlled by plant hormones, jasmonic acid, salicylic acid, and ethylene, are upregulated and others are downregulated [37–39]. Also light plays an important role, by modulating the action of plant hormones [40,41]. Other abiotic environmental factors can impact the plant-microorganism homeostasis and cause the interaction between the endophyte and its host to change from mutualistic to neutral or even parasitic [42]. This often leads to alterations in the colonisation of the plant with symbiotic microorganisms. Abiotic factors such as nutrient limitation can cause the plant to lose its ability to control microorganism development within its tissues resulting in pathogenesis. The role of spectral properties of light, especially the role of UV in the interaction is poorly examined.

Endophytic fungi are a group of symbiotic microorganisms that reside inside plant tissues [43]. They improve growth, water, and nutrient acquisition, as well as protect against stress [44]. This group of symbionts is important for the *Brassicaceae*, in which the establishment of mycorrhizal symbioses is impaired [45]. Our study aimed to understand how the presence of UV-A in the range of 350–400 nm (UV-A1) affects host-endophyte interactions in a model system. The ratio of UV-A1 to the photosynthetically active radiation (PAR) was kept close to the ratio present in sunlight. *Arabidopsis thaliana* and endophytic fungi belonging to different taxonomical units and trophic characteristics were used as models. The plants were inoculated with endophytic fungi isolated previously from wild-grown sand rock-cress *Arabidopsis arenosa* [46]. Taxonomically (Fig. 1. I), the endophytes used in this study represented three different fungal phyla: *Ascomycota* (*Phomopsis columnaris, Diaporthe eres, Paraphoma chrysanthemicola*), *Basidiomycota* (*Sporobolomyces ruberrimus*) and *Mucoromycota* (*Mucor sp.*). The representatives of *Ascomycota* belonged to *Sordariomycetes* (*P. columnaris, D. eres*) and *Dothideomycetes* (*P. chrysanthemicola)*, two classes of fungi most commonly found as plant endophytes. Although the fungi used in the experiment were isolated as endophytes from asymptomatic plant tissues [46,47], they can also act as pathogens under certain conditions. The saprotroph-pathogen continuum is a common phenomenon in the fungal lifestyle. We hypothesized that changes in light spectral properties, specifically supplementation with UV-A, would affect plant metabolism, leading to an imbalance in the plant-endophyte interaction. Consequently, we expected the relationship to shift from mutualistic or neutral to parasitic, negatively impacting plant biomass production and other biometric features.

## Materials and Methods

### Plant and endophyte growth conditions

Wild-type Col-0 *Arabidopsis thaliana, fah1–2* (N6172, ferulic acid 5-hydroxylase 1, *AT4G36220*) [48], and *tt4–11* (N2105573, chalcone synthase (CHS), SALK_020583, *AT5G13930*) [49] mutants were purchased from Nottingham Arabidopsis Stock Centre. The *tt4–11* mutant is deficient in the CHS protein, thus lacking flavonoids, with increased sinapic acid ester content [50]. The *fah1–2* mutant lacks the F5H, ferulate-5-hydroxylase, and is deficient in sinapoyl-malate and sinapoyl choline [51], required for plant defence [34] and UV-B protection [48,52]. *Arabidopsis* seeds were

surface sterilized with 70% (v/v) EtOH (Linegal Chemicals) for 5 min, 2.5% (v/v) NaOCl for 5 min, and washed 4 times with sterile distilled water. Seeds were sown *in vitro* on Petri dishes in the 1/4 MS medium without vitamins (Sigma-Aldrich) with 0.75% (w/v) sucrose and transferred to 4°C for two days. The plates were put into a custom-made environmental growth chamber. White illumination of 225 µmol $m^{-2}$ $s^{-1}$ PAR (measured through a Petri dish cover) was provided by LEDs (EPILEDs, 5000 K) supplemented with UV-A of 13 µmol $m^{-2}$ $s^{-1}$ from fluorescent tubes (Philips TL 8 W BLB) covered with either a Lee #226 gel filter (blocks UV-A and UV-B), or with a polyester film (125 µm thick, Autostat CT5, MacDermid), which blocks UV-B but passes UV-A and visible light. The applied UV-A radiation was contained in the UV-A1 waveband (350–400 nm) (S1 Fig). In our study, the ratio between photon irradiance of UV-A in the 350–400 nm waveband to PAR was close to the ratio in the standard 1.5 air mass sunlight spectrum (0.051 in ASTM G173 - 03) [53]. It was selected as an approximation of clear sky conditions at the mid-latitudes during the vegetative period. On the 8th day of cultivation, plants were transferred to the Modified Strullu Romand medium (MSR) and inoculated with one of the endophytes. Identification of fungal strains was based on the sequences of ITS DNA [54,55] described in [56,57]. The NCBI database accession numbers for the strains were *Paraphoma chrysanthemicola* - OR077971, *Phomopsis columnaris* - OR077985, *Diaporthe eres* - OR078011, *Mucor* sp. - OR078038, and *Sporobolomyces ruberrimus* - ON331994. Endophytic fungi mycelia were cultivated on Petri dishes with PDA medium (GRASO Biotech) at 25°C for 14 days in darkness in a standard incubator. Plant roots were inoculated using plugs of the approximate size of 3x3 mm placed between the roots with the mycelium oriented towards the medium for *Paraphoma chrysanthemicola*, *Phomopsis columnaris*, and *Diaporthe eres*. *Mucor sp.* spores were picked from 25 $mm^2$ of the plate (concentration approximately 0.16–0.18 x $10^6$ $mm^{-2}$ on the plate) with an inoculation loop and placed between the roots of *A. thaliana* plants. The yeast cell culture of *Sporobolomyces ruberrimus* ($OD_{600}$ approximately 7.0) was spread directly on the whole root surface with an inoculation loop. Mock-inoculated plants served as a control. Plants were cultivated for the next 10 days. The plant condition was then assessed by measuring wet shoot and root weight. Root morphology parameters were acquired and analysed using WinRHIZO *Arabidopsis* software (Regent Instruments Inc). The total length of the root system, average root diameter, and total root volume were measured. For the experiment with the wild type, *tt4,* and *fah1* mutants, the main root length was marked every 12 h after inoculation for 5 consequent days and analysed with ImageJ.

## Construction of a phylogenetic tree

The phylogeny was inferred using the Maximum Likelihood method and Tamura-Nei (1993) model [58] of nucleotide substitutions, and the tree with the highest log likelihood (-20 328.52) is shown. The initial tree for the heuristic search was selected by choosing the tree with the superior log-likelihood between a Neighbor-Joining (NJ) tree [59] and a Maximum Parsimony (MP) tree. The NJ tree was generated using a matrix of pairwise distances computed using the p-distance [60]. The MP tree had the shortest length among 10 MP tree searches, each performed with a randomly generated starting tree. The analytical procedure encompassed 25 coding nucleotide sequences using 1st, 2nd, 3rd, and non-coding positions with 2577 positions in the final dataset. Evolutionary analyses were conducted in MEGA12 [61] utilizing up to 8 parallel computing threads.

## *Arabidopsis* pot cultures

Wild-type *Arabidopsis* seeds were sown on Jiffy-7 peat pellets and then stratified at 4°C for 2 days. Plants were grown in a cultivation chamber in the same conditions as *in vitro* cultures. When the plants had developed the first pair of leaves (on the 9th day), they were transplanted into a pot with a substrate of perlite: sand 1:1 (v/v). Plants in the perlite: sand medium were watered with 30 ml of Hogland's medium (Sigma-Aldrich), 3 times a week. Liquid cultures of *S. ruberrimus* were carried out for 3 days at 28°C with shaking at 120 rpm in 100 ml flasks with 30 ml of 2% (w/v) malt extract. On 3rd day, optical density at 600 nm reached approximately 7.0. Then 90 ml of the yeast culture was centrifuged at 4500 g for 10 min at room temperature. The resulting pellet was washed 3 times by suspending it in 30 ml of 0.9% (w/v) NaCl and centrifuging

at 4500 g for 10 min at room temperature. Yeast culture was finally suspended in 200 ml of 0.9%(w/v) NaCl and used for watering the plants. Plants were inoculated with 10 ml of the suspension of *S. ruberrimus* on the 12[th] and 19[th] day of development, i.e., on the 3[rd] and 10[th] day after transplanting into the new medium. The control, mock-inoculated plants were watered with 10 ml of 0.9% (w/v) NaCl solution.

## Anthocyanin, flavonol, and chlorophyll content analysis

For anthocyanin content estimation, the protocol was adapted from [62]. Shoot samples were grounded to powder in liquid nitrogen, suspended in 80% (v/v) methanol solution in proportion 1:10 (mass/volume), and homogenized in TissueLyser II (Qiagen) with glass beads at the oscillation frequency of 30 Hz for 5 min. TissueLyser II adapters were precooled at -20°C for 24 h. Samples were centrifuged at 4°C in 14000 g for 15 min. 250 µl of each sample was resuspended in 4.5 ml of 2% (v/v) HCl solution in water and 250 µl of 1% (v/v) HCl solution in 95% (v/v) ethanol and mixed by inverting. After incubation in the darkness for 15 min, the anthocyanin content was measured spectroscopically at 532 nm in three independent technical replicates. The concentration was estimated using a calibration curve prepared for malvidin 3-glucoside [62]. Flavonols and chlorophyll contents were measured in the upper epidermis of the 6[th], 7[th], and 8[th] intact *Arabidopsis* leaf using the Dualex Scientific fluorimeter (Force-A).

## Laser scanning confocal microscopy

The presence of the endophyte was analysed by confocal microscopy using WGA-Texas Red stained fungi mycelium as in [63]. Root and leaf fragments were cleared in 10% (w/v) KOH for 30 min, then rinsed in sterile water for 5 min. The tissues were incubated for 30 min with 20 µg/ml WGA-Texas Red (a stock solution of WGA-Texas Red 1 mg/ml in 0.01 M Phosphate, 0.15 M NaCl, 0.05% (w/v) sodium azide) and then washed three times for 10 min in the phosphate buffer. The samples were observed using the Axio Observer.Z1 inverted microscope (Carl Zeiss, Jena) equipped with the LSM 880 confocal module. Plan-Neofluar 40x objective (NA 1.3) was used with oil immersion. Texas Red fluorescence was excited with the 594 nm He-Ne laser. Emission in the range of 599 nm – 630 nm was recorded as the red channel.

## qPCR -based quantification of endophytic fungi

To quantify the amount of each fungus, a qPCR method was employed to detect endophyte-specific fungal translational elongation factor 1α (tef1α) or chitinase (*S. ruberrimus*) DNA. The roots and leaves of *A. thaliana* were surface sterilized. Plant organs were rinsed three times with sterile distilled water, incubated in 0.5% (v/v) sodium hypochlorite solution (NaOCl) (Sigma-Aldrich) with Tween20 (Bio-Shop) for 1 min, rinsed with sterile distilled water for 2 min, and incubated in 75% (v/v) ethanol solution for 30 s. Then the plant material was rinsed three times with sterile water (1 min, 5 min, 1 min). Air-dried samples were immediately frozen in liquid nitrogen and kept at -80°C. Root and shoot samples were grounded separately in liquid nitrogen. DNA extraction was performed according to [64] modified by [65]. DNA was extracted using 700 µl of 2% (w/v) CTAB solution (Sigma-Aldrich) in 0.1 M TRIS-HCl pH 8.0, 1.4 M NaCl, 0.02 M EDTA, 1% PVP with 1.5 µl of β-mercaptoethanol (Sigma-Aldrich). Samples were vortexed vigorously and incubated with shaking at 65°C for 60 min. 5 µl of RNAse solution was added to remove RNA and incubated for 10 min at room temperature. The samples were centrifuged at room temperature for 5 min at 15000 g. The supernatants were transferred to a new Eppendorf tube containing 700 µl of chloroform and centrifuged for 30 min at 15000 g at room temperature. To precipitate DNA, the upper aqueous phase was transferred to a new Eppendorf tube with 600 µl of 70% (v/v) ice-cold isopropanol solution (-20ºC), and incubated overnight at -80°C. The following day, the samples were thawed on ice and centrifuged for 40 min at 15000 g at 4°C. The supernatants were removed, and the DNA pellets were washed with 70% (v/v) ethanol, then 96% ethanol, and centrifuged for 5 min at 15000 g. The DNA pellet was re-suspended in 30 µl of TE buffer and incubated with mixing for 1 h at 37°C. Then 0.1 volume of 3 M sodium acetate (Sigma-Aldrich) and 2.5 volume of 70% (v/v) ice-cold isopropanol solution were added to the DNA samples. After overnight incubation at -80°C, the samples were thawed on

ice, centrifuged for 40 min at 15000 g at 4°C, washed with 70% (v/v) ethanol, then with 96% (v/v) ethanol, and centrifuged for 5 min at 15000 g. The DNA pellet was re-suspended in 30 μl of the TE buffer pH 8.0 (Invitrogen), and incubated with shaking for 1 h at 37°C. qPCR was performed with a SYBR Green JumpStart Taq Ready Mix (Sigma-Aldrich) and a Biorad Touch CFX96 Real-Time PCR Detection System. 20 ng of total DNA extracted from samples and 0.33 μM of each primer from the pair were used for a single reaction, performed in triplicates. The primers are listed in the Table S1 of the S1 Appendix. After initial denaturation at 95°C for 4 min, 50 cycles of denaturation at 95°C for 10 s, followed by annealing at the temperature specific to the respective primer for 15 s, and elongation at 72°C for 20 s were performed.

### Real-time PCR

RNA was isolated from above-ground plant parts. RNA isolation and real-time PCR were performed according to [66], except that oligodT primers were used for RNA reverse transcription. Primer sequences are listed in the Table S2 of the S1 Appendix. Each sample was run in three technical replicates. The relative expression of each gene in a sample was determined using the mean value of $Ct$ for all samples. For reference genes, the levels of $SAND$ and $PDF2$ were quantified, based on [67]. Expression levels were normalized using factors calculated by geNorm v3.4 [68].

### Lipid peroxidation measurements

The leaf rosettes were grounded to a powder in a mortar in liquid nitrogen without allowing the tissue to thaw. Each sample was divided into 3 technical replicates of 100 mg. If the weight of the sample was too low, 50 mg of grated plant tissue was used. The sample was suspended in a 0.1% (v/v) aqueous solution of trichloroacetic acid in a ratio of 1:10 and homogenized using glass beads in a TissueLyser II (Qiagen) for 10 min at 30 Hz. Before use, the TissueLyser II block was cooled at -20°C for at least 24 h. The sample was centrifuged at 10 000 g at 4°C for 5 min. After centrifugation, the supernatant was removed and mixed by pipetting with freshly prepared 0.5% (w/v) thiobarbituric acid (TBA) dissolved in 20% (w/v) trichloroacetic acid. One part of the supernatant was mixed with 4 parts of the TBA solution. Samples were then incubated in a thermoblock at 95°C for 30 min. Samples were chilled on ice and then centrifuged at 10000 g at 4°C for 5 min. The concentration of malondialdehyde (MDA), a lipid peroxidation marker, was measured using the reaction of TBA with MDA, which yields coloured MDA-$(TBA)_2$ adduct as in [69]. The adduct concentration was determined spectrophotometrically by measuring the absorbance of the sample at the wavelength of 532 nm in two chemical replicates.

### Measurement of the photosynthetic parameters of leaves

Photosynthetic parameters were analysed in the 6th and 7th rosette leaf using Dual-PAM-100 fluorimeter (Heinz Walz GmbH). To determine the slow kinetics with the induction curve of chlorophyll, whole plants were dark-adapted for 30 min before the measurement. The $F_v/F_m$ ratio was determined using weak measuring light ($F_0$) and a saturating pulse ($F_m$) of 10000 μmol $m^{-2}$ $s^{-1}$ for 600 ms. After a 50 s pause, red actinic light (AL) of 75 μmol $m^{-2}$ $s^{-1}$ PAR was turned on. Then sequential saturation pulses (SP) of 10000 μmol $m^{-2}$ $s^{-1}$ were applied, which lasted 600 ms at 20 s intervals. Light curves were measured on irradiated plants. Leaves were irradiated with red actinic light (AL) of increasing irradiance of 0, 11, 18, 27, 58, 100, 131, 221, 344, 536, 830 μmol $m^{-2}$ $s^{-1}$, with each phase lasting 30 s. Pulses of saturating light lasted 600 ms and were applied every 30 s.

### Statistical analysis

The position of plates and pots with plants in the cultivation chamber was randomized. Plates were repositioned daily to reduce systematic errors due to inhomogeneity in growth conditions in the chamber. Statistical analysis was performed using the R software [70]. No data points were rejected as outliers. The details of the analysis and the obtained results are provided in the S2 Appendix. To avoid pseudoreplication, when measurements of the same property were obtained for multiple plants growing on the same plate, they were averaged before analysis and the plate means were treated as

single replicates. Linear models were fitted using the *nlme* package [71,72], with the *gls* (when all factors were fixed) or *lme* command (when a random factor corresponding to a batch of plants was included), with the interaction between fixed factors. In each case, the linear models allowed for unequal variances between groups. Preliminary tests of variance homogeneity were not used to avoid distortion of the significance levels [73]. As this work explores whether the presence of UV-A modulates the effect of endophyte inoculation, the significance of interaction terms, as examined with ANOVA, was important for the interpretation of the results. Type III ANOVA was calculated. In type III ANOVA, the significance of a main effect is examined comparing the full model with a reduced model, in which the main effect of the factor of interest is removed, but its interactions are retained. This approach has been suggested to be more appropriate when substantial interactions are likely [74] which is often the case in studies involving stress factors [75]. However, as the number of replicates was similar in all groups, the choice of ANOVA type did not substantially affect the results of the analysis. The analysis of the qPCR results and the gene expression levels was performed on log-transformed data, other data were not transformed. Differences between cell means of responses or log-transformed responses were examined using the *emmeans* package [76]. The approximate number of degrees of freedom was calculated with Satterthwaite's method, which is designed for data in which variance may differ between groups. The Benjamin-Hochberg or Hommel method was used to adjust *p* values for cell mean differences, as specified in the S2 Appendix.

## Results

### Colonisation of plants by endophytes

To examine whether the UV-A radiation affects the colonisation of plant tissues by endophytic fungi, *Arabidopsis thaliana* wild-type seedlings were grown under the illumination of ca. 225 µmol m$^{-2}$ s$^{-1}$ PAR with or without supplementation of 13 µmol m$^{-2}$ s$^{-1}$ UV-A. After eight days, plants grown in the presence of UV-A showed reduced root length compared to those grown only under PAR (S2 Fig., S2 Appendix, Table S5). At this time point the plants were inoculated with *Paraphoma chrysanthemicola* (Fig.1. a), *Phomopsis columnaris* (Fig. 1. b), *Diaporthe eres* (Fig. 1. c), *Mucor* sp. (Fig. 1. d), *Sporobolomyces ruberrimus* (Fig. 1. e), and cultivated in the Modified Strullu Romand medium (MSR) for ten days (S3 Fig.). Plants that were not inoculated served as a control. Plant colonisation with the endophytes was assessed separately in shoots and roots.

The localization of endophytes within plant tissues was analysed using Wheat Germ Agglutinin lectin conjugated with Texas Red. The fluorescently labelled lectin selectively binds to chitin, the main polysaccharide of the fungal cell walls. Fungal mycelia were observed only in roots inoculated with *Mucor* sp.*, Paraphoma chrysanthemicola, Diaporthe eres*, and *Phomopsis columnaris* (Fig. 1g,h,i,j, S4 Fig.) regardless of the light conditions during the growth of plants. However, yeast cells of *Sporobolomyces ruberrimus* were present in the roots and shoots of inoculated plants (compare Fig.1k and S4 Fig.). A non-specific signal was observed in control, mock-inoculated samples (Fig. 1f, S4 Fig.). Further, to confirm and quantify the fungal abundance in plant tissues, the qPCR-based method was utilised. The presence of fungal chitinase (for *S. ruberrimus*) or translational elongation factor 1α (tef1α, for other fungi) genes in DNA extracted from surface-sterilized plants was investigated (Fig. 2). In the case of *Paraphoma chrysanthemicola, Phomopsis columnaris,* and *Mucor* sp., the fungus was found in inoculated roots (Fig. 2a,b,d). Colonisation by *Diaphorte eres* and the yeast *Sporobolomyces ruberrimus* was detected both in shoots and roots (Fig. 2c,e). The effect of interaction between the inoculation and light conditions on qPCR values was not significant for any of the studied endophytes (ANOVA results in the S2 Appendix, Table S1), indicating that the presence of UV-A does not substantially influence endophyte colonisation.

### Plant growth parameters

To investigate the interaction between the effect of UV-A presence (two levels, UV-A supplementation or visible light only) and the effect of endophyte inoculation (six levels, including the mock-inoculated control) on the host plant growth, we

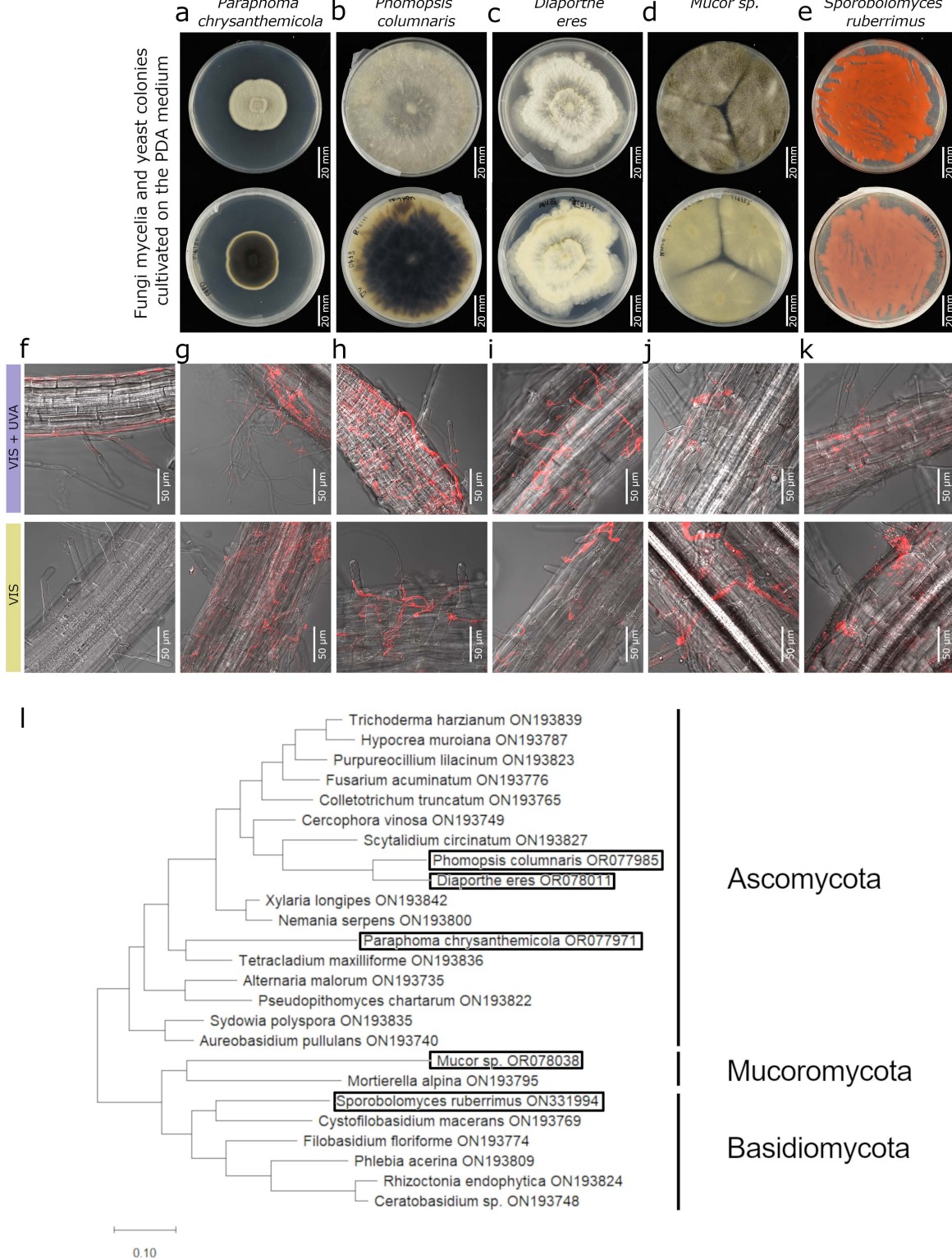

**Fig. 1. Fungi mycelia used in the study, cultivated on the PDA medium** (a) *Paraphoma chrysanthemicola,* (b) *Phomopsis columnaris,* (c) *Diaporthe eres*, (d) *Mucor* sp., and yeasts (e) *Sporobolomyces ruberrimus.* The top view is in the upper row, the bottom view is in the lower row. Roots

of *Arabidopsis* mock-inoculated (f) or inoculated with endophytes *Paraphoma chrysanthemicola* **(g)**, *Phomopsis columnaris* **(h)**, *Diaporthe eres* **(i)**, *Mucor* sp. **(j)**, *Sporobolomyces ruberrimus* **(k)**, grown in the presence or absence of UV-A radiation, stained with Wheat Germ Agglutinin lectin conjugated with Texas Red. Laser-scanning confocal images show the transmitted light channel merged with the channel of red fluorescence (599–630 nm). Plants not inoculated with the endophyte serve as a control for natural autofluorescence. Scale bars = 50 μm. **(l)** A phylogenetic tree with the highest log likelihood inferred using the Maximum Likelihood method and Tamura-Nei model of nucleotide substitutions. The scale bar shows the length of the branch representing the amount of genetic change of 0.10.

examined the selected plant growth parameters in plants in all combinations of the levels of both factors. The presence of UV-A radiation resulted in shoot growth inhibition in mock-inoculated control plants as well as in inoculated plants (regardless of endophytic species) (Fig. 3a). The effects of endophyte inoculation and UV-A radiation on the fresh weight of shoots were both significant ($F_{(5,91)}$ = 9.82, $p = 2.5 \cdot 10^{-14}$; $F_{(1,91)}$ = 81.93, $p = 2.5 \cdot 10^{-14}$, respectively), though their interaction was insignificant ($F_{(5,91)}$ = 0.53, $p = 0.75$) (ANOVA results in the S2 Appendix, Table S2). The fresh shoot weight was significantly lower in plants supplemented with UV-A than in plants grown only in visible light, irrespective of the type of inoculation (Fig. 3. a, $p$ values in the S2 Appendix, Table S2). Inoculation with *Mucor* sp. and *Paraphoma chrysanthemicola* resulted in a decrease in the shoot weight, in plants grown with and without UV-A. To further characterise the impact of UV-A on the aerial part of the plants, the anthocyanin content was assessed by the spectrophotometric method (Fig. 3. b). Anthocyanin elevation may result from UV irradiation, but also other stress factors, such as the presence of endophytes. The UV-A supplementation affected the anthocyanin levels in plants ($F_{(1,82)}$ = 22.4, $p = 9.1 \cdot 10^{-6}$). However, the effect of the endophyte inoculation ($F_{(5,82)}$ = 32.24, $p = 0.058$) and the interaction effect ($F_{(1,87)}$ = 1.11, $p = 0.36$) were not significant. While the mean anthocyanin content in our sample was greater for all endophytes in plants acclimated to UV-A than those grown in visible light alone, the differences were not statistically significant (Fig. 3. b). On the day of inoculation, plants acclimated to UV-A exhibited shorter roots than those grown only in visible light (S2 Fig.), but the effect of light conditions was not significant after the additional ten days of growth in the presence of fungi. The presence of endophyte affected fresh root weight ($F_{(5,91)}$ = 2.8, $p = 0.021$) (Fig. 3. c), root system length ($F_{(5,90)}$ = 7.8, $p = 4 \cdot 10^{-6}$) (Fig. 3. d), average root volume ($F_{(5,90)}$ = 6.2, $p = 7.1 \cdot 10^{-5}$) (Fig. 3. e), and average root diameter ($F_{(5,90)}$ = 18.5, $p = 1.3 \cdot 10^{-12}$) (Fig. 3. f). However, the interaction of light conditions and the endophyte presence was nonsignificant for the root fresh weight, length, and volume. The means of measured root system length and root volume in our sample were in general smaller for inoculated plants than for the control, though only in some cases were the differences significant. In particular, when UV – supplementation was applied, both average root volume and the length of the root system were significantly smaller in plants inoculated with *Diaporthe eres* than in the mock-inoculated plants ($p = 0.013$ and $p = 0.00066$ for volume and length, respectively) (Fig. 3d,e). The sample means of average root diameter (Fig. 3. f) were in almost all cases greater in plants inoculated with the endophytes than in the mock-inoculated ones, with the difference being significant for *Paraphoma chrysanthemicola*, *Diaporthe eres*, and *Sporobolomyces ruberrimus* in both types of light conditions ($p$ values in the S2 Appendix, Table S2).

### Expression of genes involved in plant defence and stress in *Arabidopsis* above-ground plant parts

Plant-microorganism interactions necessitate adaptations in plant secondary metabolism, including the activation of phenolic compound production [77] and the fine-tuning of defence related hormone homeostasis. A well-documented aspect of this adaptation is the down-regulation of salicylic acid-related plant defence in mutualistic plant-microorganism interactions [39]. To investigate the influence of UV-A radiation on plant responses to endophytes, we examined the expression levels of genes involved in the synthesis of flavonoids, [78] *PHENYLALANINE AMMONIA-LYASE 1 (PAL1)* and *CHALCONE SYNTHASE (CHS)* [78], both of which are known to be induced by UV radiation. The effect of endophyte inoculation on *CHS* transcript, as tested with ANOVA, was significant ($F_{(1,96)}$ = 3.28, $p = 0.009$), though the differences in means between particular groups were not significant (Fig. 4. A, $p$ values in the S2 Appendix, Table S3). PAL1 and ICS1 are involved in different pathways of salicylic acid biosynthesis, contributing to local and systemic acquired resistance [79].

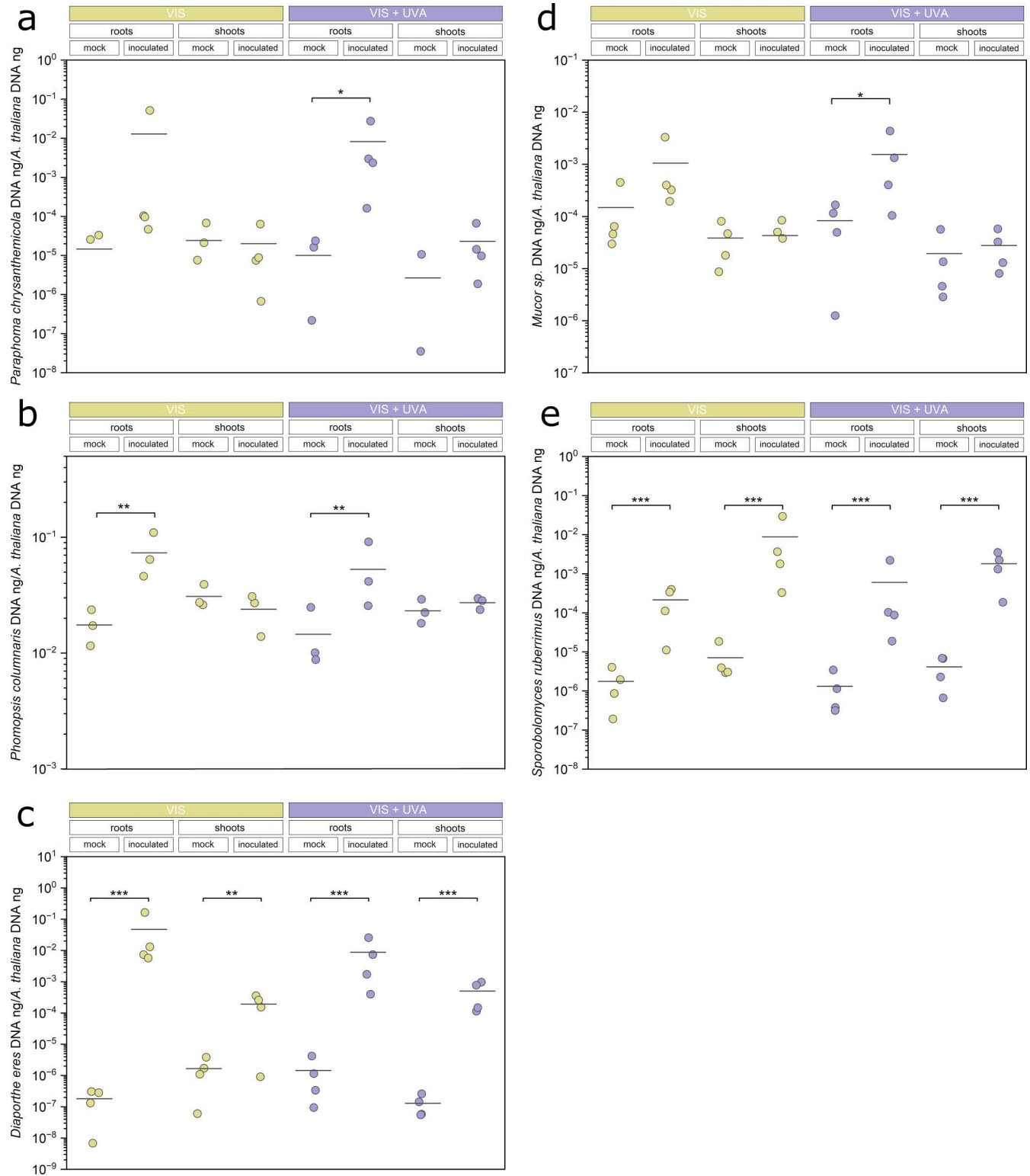

**Fig 2. Endophyte inoculation of *Arabidopsis thaliana*, grown in the presence or absence of UV-A, assessed by the qPCR method.** The ratio of fungal DNA to plant DNA in roots and shoots of plants inoculated with *Paraphoma chrysanthemicola* **(a)**, *Phomopsis columnaris* **(b)**, *Diaporthe eres* **(c)**, *Mucor sp.* **(d)**, *Sporobolomyces ruberrimus* **(e)**. The experiment was performed in four biological replicates. Brackets show the significant differences in

means of log-transformed qPCR measurements between inoculated and mock-inoculated samples, from the same organ and light conditions (* $0.01 < p \leq 0.05$, ** $0.001 < p \leq 0.01$, *** $p \leq 0.001$, adjusted for multiple comparisons with Hommel's method). The set of all tested contrasts is in the S2 Appendix, Table S1.

The expression of both genes was influenced by the endophyte inoculation ($F_{(5,96)}$ = 4.32, $p = 0.0014$ for *PAL1*; $F_{(5,96)}$ = 11.5, $p = 9.7 \cdot 10^{-9}$ for *ICS1*) and UV-A ($F_{(1,96)}$ = 5.75, $p = 0.018$ for *PAL1*; $F_{(1,96)}$ = 9.48, $p = 0.0027$ for *ICS1*), though the interactions of these factors were not significant ($p = 0.86$ for *PAL1*; $p = 0.62$ for *ICS1*) (Fig. 4b,c). In most cases, the differences between means for particular groups were not significant and were small in magnitude. The presence of *Paraphoma chrysanthemicola* and *Sporobolomyces ruberrimus* increased *ICS1* levels regardless of light conditions (Fig. 4. c). *PDF1.2*, an ethylene- and jasmonate-responsive plant defensin, is upregulated upon fungal infection of leaves [80]. We observed that the *PDF1.2* transcript level was affected by the presence of the endophytes ($F_{(5,96)}$ = 12.1, $p = 4.5 \cdot 10^{-9}$), moreover, the influence of the endophyte was modified by the UV-A supplementation ($F_{(5,96)}$ = 2.93, $p = 0.017$ for the interaction term). In particular, the presence of *Paraphoma chrysanthemicola, Diaphorte eres*, and *Sporobolomyces ruberrimus* significantly elevated the *PDF1.2* transcript levels as compared to the mock-inoculated control only when light was supplemented with UV-A (Fig. 4. d; *p* values in the S2 Appendix, Table S3).

### Expression of genes involved in short-wavelength radiation signalling

We also investigated whether biotic factors may affect, similarly to light, the expression of photoreceptor genes and in consequence alter plant physiology. We examined whether the presence of endophyte affects the expression levels of genes encoding plant photoreceptors sensitive to the UV-A range (S5 Fig.; *p* values in the S2 Appendix, Table S3). The main effects of the endophyte inoculation ($F_{(5,96)}$ = 2.83, $p = 0.02$) and the UV-presence ($F_{(1,96)}$ = 4.48, $p = 0.037$) were significant for the *PHOT2* gene, but the interaction effect was not ($p = 0.401$), while the main effect of UV-A was significant for *PHOT1* ($p = 0.033$). No changes in relative transcript levels were observed for *CRY1, CRY2,* and *UVR8* genes (S5 Fig.; *p* values in the S2 Appendix, Table S3).

### Physiological parameters of *Arabidopsis* grown in pot cultures inoculated with *S. ruberrimus*

We detected *Sporobolomyces ruberrimus* in shoots, using both microscopic examination and qPCR. To determine whether the UV-A treatment affects the influence of the endophyte on host photosynthetic efficiency, we performed experiments in pot cultures, measuring several photosynthetic parameters in mature *Arabidopsis* leaves grown in only visible light or acclimated to UV-A radiation, in the same conditions as the *in vitro* experiment (Fig. 5, *p* values in the S2 Appendix, Table S4). *Arabidopsis* plants cultivated in pots filled with perlite and sand substrates were inoculated twice with the yeast suspension. After 26 days of growth, flavonol (Fig. 5. b) contents were affected by UV-A in a manner dependent on the presence of *S. ruberrimus* ($F_{(1,69)}$ = 6.32, $p = 0.014$ for the interaction effect). Lipid peroxidation was assessed with the MDA (malondialdehyde) assay, measuring the concentration of the final product of lipid peroxidation. Neither the presence of *S. ruberrimus* nor UV-A changed MDA levels, (Fig. 5. c). We also used a Dual Pam Fluorimeter in a saturation pulse mode, to register time-dependent changes in chlorophyll fluorescence of *Arabidopsis* plants inoculated and mock-inoculated with *S. ruberrimus* grown with or without UV-A supplementation. No substantial differences in the photosynthetic efficiency parameters: Y(II), ETR, and NPQ of leaves were observed for induction (Fig. 5d-f) and light (Fig. 5g-i) curves.

### Inoculation of phenolic pathway mutants with *S. ruberrimus* and *P. chrysanthemicola*

To understand the role of flavonoid accumulation in colonisation by *S. rubberrimus*, we used the *tt4–11* mutant deficient in the CHS protein, which lacks flavonoids but shows increased sinapic acid esters [50]. Given the role of sinapate esters in plant defence [34] and UV-B protection [48,52], we also examined the *fah1–2* mutant, which lacks the F5H,

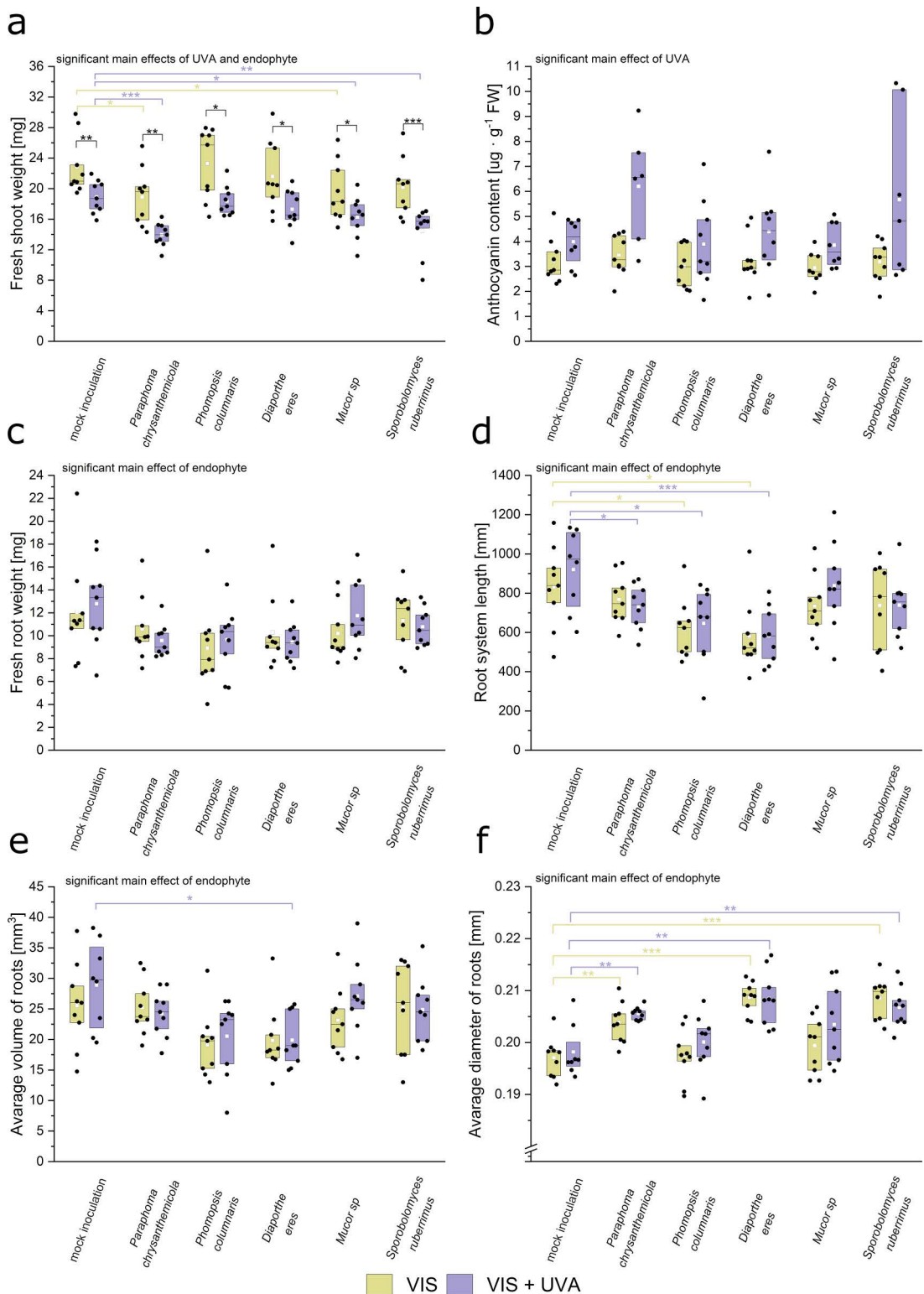

**Fig. 3. Fresh weight of shoots (a), shoot anthocyanin content (b), fresh weight of roots (c), root system length (d), average root volume (e), and average root diameter (f) of 18-day-old plants, either mock-inoculated (control) or inoculated with *Paraphoma chrysanthemicola*,**

*Phomopsis columnaris, Diaporthe eres, Mucor* sp.*, Sporobolomyces ruberrimus.* Plants were grown in the presence or absence of UV-A. The experiment was performed in 9 biological replicates (plates) for every combination of inoculation type and light conditions. Black brackets show the significant differences in means between plants inoculated with the same endophyte, but subject to different light conditions. Yellow (VIS) and violet (VIS + UV) brackets show significant differences in means between inoculated samples and the mock-inoculated control (* $0.01 < p ≤ 0.05$, ** $0.001 < p ≤ 0.01$, *** $p ≤ 0.001$, adjusted for multiple comparisons with the Benjamin-Hochberg method). The set of all tested contrasts is in the S2 Appendix, Table S2.

ferulate-5-hydroxylase, a cytochrome P450-dependent monooxygenase, and, in consequence, is deficient in sinapoyl-lmalate and sinapoyl choline [51]. We also examined *P. chrysanthemicola* as a mycelium-forming fungus that affected the growth of plants in our experiments. The qPCR analysis of the amount of endophytes (Fig. 6a, b, the S2 Appendix, Table S6) did not reveal any differences between plant lines in their susceptibility to colonization, as evidenced by the non-significant effect of interaction between the plant line and inoculation effect on the qPCR readings ($F_{(2,22)} = 0.12$, $p = 0.89$ for *P. chrysanthemicola*; $F_{(2,22)} = 0.65$, $p = 0.53$ for *S. rubberrimus*). Likewise, we did not detect a significant effect of the presence of UV-A on colonisation ($F_{(1,22)} = 1.60$, $p = 0.22$ for *P. chrysanthemicola*; $F_{(1,22)} = 0.01$, $p = 0.94$ for *S. rubberrimus*). Fresh shoot weight (Fig. 6. C, the S2 Appendix, Table S7) depended on the plant line ($F_{(2,121)} = 8.59$, $p < 0.001$) and was affected both by the presence of endophytes ($F_{(2,121)} = 46.2$, $p < 0.001$) and UV-A ($F_{(1,121)} = 10.8$, $p = 0.0014$). However, no interaction between these factors was found. Lower shoot weight was recorded in plants inoculated with *P. chrysanthemicola* and *S. rubberrimus* than in mock-inoculated ones, regardless of the plant genotype and light treatment. The differences were statistically significant in most cases, as specified in the S2 Appendix, Table S7. Fresh root weight was affected by the presence of endophytes ($F_{(2,121)} = 72.62$, $p < 0.001$) and, to a lesser extent, was dependent on plant line ($F_{(2,121)} = 18.69$, $p < 0.001$) (Fig. 6. d). Neither the effect of UV nor interactions between factors were significant. Seedlings inoculated with either of the endophytes exhibited smaller root weights than mock-inoculated ones. In most cases, these differences were statistically significant and substantial in magnitude (S2 Appendix, Table S7). We also analysed the main root elongation during the first five days after inoculation with the endophyte (S6 Fig.). The main root length 100 h after the inoculation was significantly affected by the presence of endophytes ($F_{(2,123)} = 176.3$, $p < 0.001$), UV-A ($F_{(1,123)} = 30.8$, $p < 0.001$) and by the plant line ($F_{(2,123)} = 3.46$, $p = 0.03$) (Fig. 6. e). Notably, the effect of UV-A supplementation depended on the presence of endophyte, as indicated by the significant interaction term ($F_{(2,123)} = 5.95$, $p = 0.003$). In plants inoculated with *P. chrysanthemicola,* a UV-A-induced decrease in main root length was observed in the wild-type and *tt4–11* mutant, but not in the *fah1–2* plants. Inoculation with *S. rubberrimus* resulted in a much shorter main root than in the control conditions, regardless of the genotype and light treatment.

## Discussion

Few studies have investigated crosstalk between short-wavelength radiation and colonisation by microorganisms. The presence of UV-B enables root colonisation by *Deinococcus* in *Nicotiana attenuata* through the action of UVR8 and chalcone synthase [81]. Endophytes and elevated UV-A and UV-B radiation modulate herbivore susceptibility and the growth of grasses, *Lolium perenne*, and *Festuca spp.* [82]. In an Antarctic dicot, *Colobanthus quitensis* the presence of endophytes improves UV tolerance by reducing lipid peroxidation [83] and increases biomass production by modulating plant hormone synthesis [33]. UV-B radiation increases resistance to plant pathogenic fungus *Botrytis cinerea* in *Arabidopsis thaliana* through the action of UVR8, which leads to the accumulation of phenolic compounds such as sinapate [34]. UV-B radiation affects flavonoid content in *Fagopyrum* colonized with fungi [35]. UV-A radiation improves tolerance against *Fusarium oxysporum* in tomato [36].

Our data showed that UV-A, applied in the 350–400 nm range, reduced the fresh weight of *Arabidopsis* shoots regardless of endophyte presence. UV-A has a positive impact on the rosette diameter of several *Arabidopsis* ecotypes grown in soil [5]. It also elevates the area and blade length of *Arabidopsis* Columbia (Col-0) mature leaves grown in soil culture.

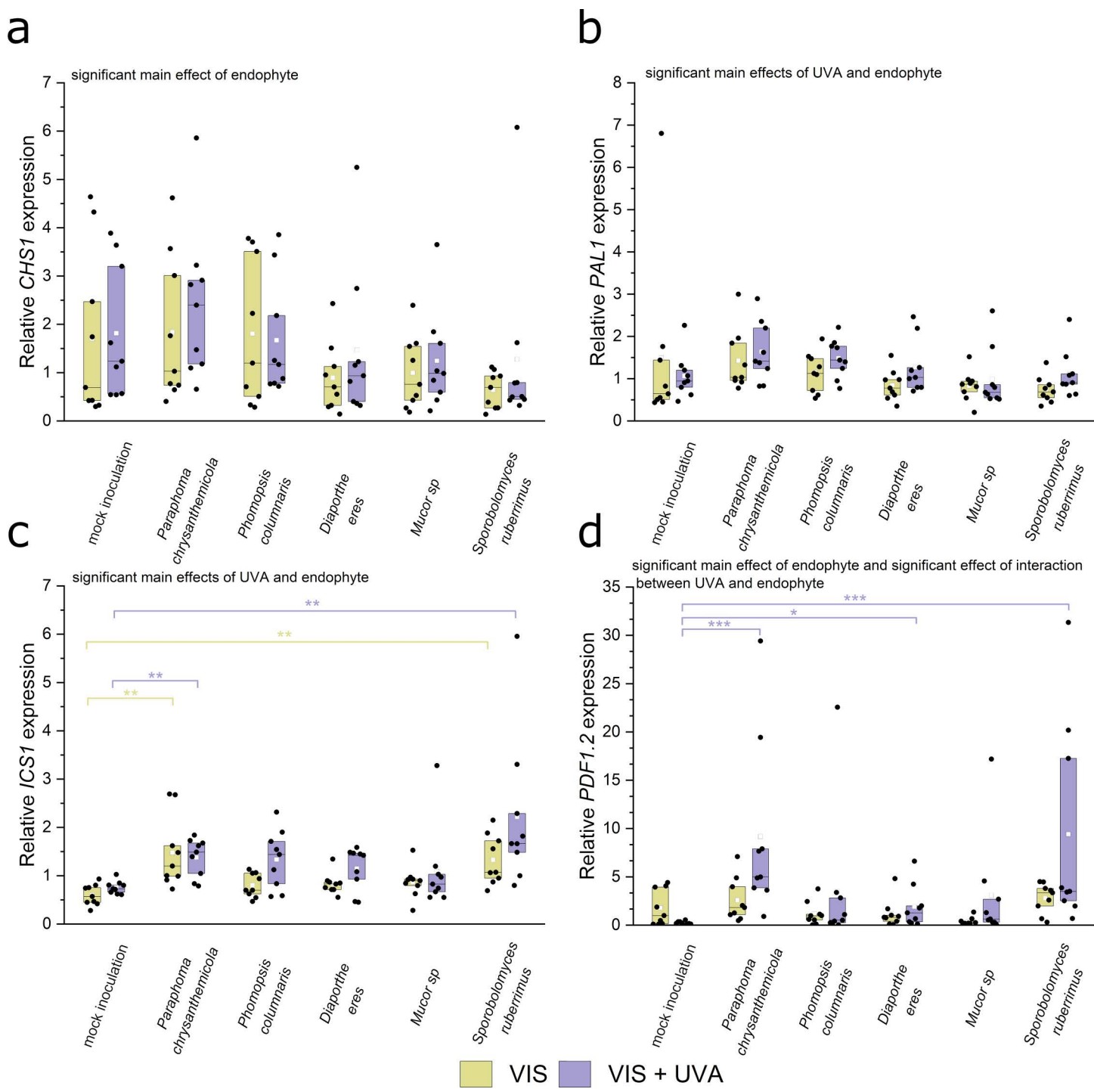

**Fig. 4. Relative transcript levels of *CHS* (a), *PAL1* (b), *ICS1* (c), and *PDF1.2* (d) genes in leaves of 18-day-old plants, either mock-inoculated (control) or inoculated with *Paraphoma chrysanthemicola*, *Phomopsis columnaris*, *Diaporthe eres, Mucor* sp*., Sporobolomyces ruberrimus*.** Plants were grown in the presence or absence of UV-A. The experiment was performed in 9 biological replicates. Yellow (VIS) and violet (VIS + UV) brackets show significant differences in means between inoculated samples and the mock-inoculated control (* $0.01 < p \leq 0.05$, ** $0.001 < p \leq 0.01$, *** $p \leq 0.001$, adjusted for multiple comparisons with the Benjamin-Hochberg method). The set of all tested contrasts is in the S2 Appendix, Table S3.

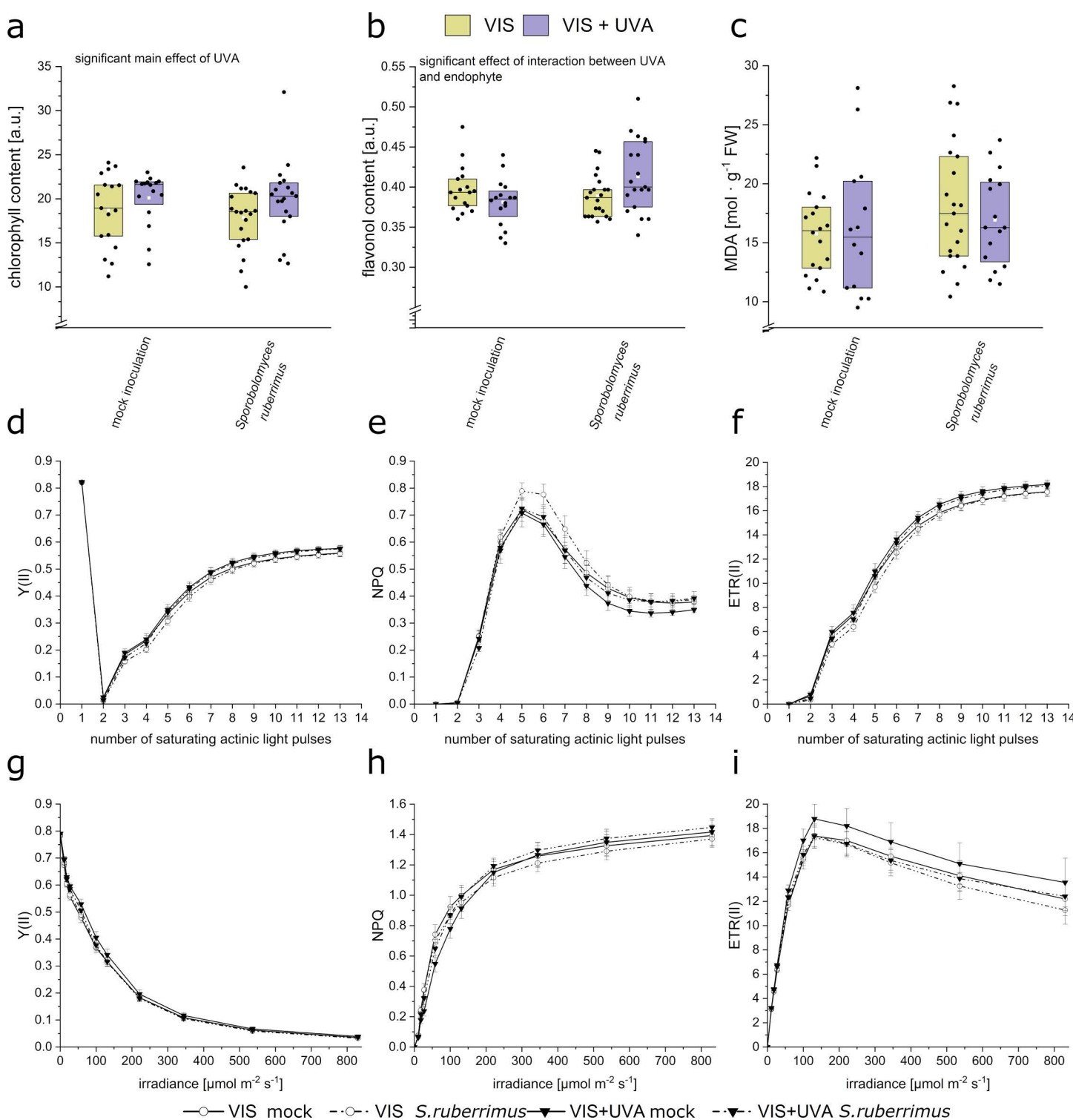

**Fig. 5. Chlorophyll content (a), flavonol content (b), MDA (c) and photosynthetic parameters (d-i) in *Arabidopsis thaliana* pot cultures inoculated with *Sporobolomyces ruberrimus* grown in the presence or absence of UV-A. (a - c)** The experiment was repeated six times. Each dot represents a measurement obtained from a single plant (in a – b, the mean of values recorded for the 6th, 7th, and 8th leaf). The results of the statistical analysis are in the S2 Appendix, Table S4. **(d - f)** Induction curves of *Arabidopsis* pot cultures inoculated with *S. ruberrimus* with standard error. d – Y(II),

e – NPQ, f – ETR. Dark-adapted plants were irradiated with the initial Saturating Pulse (SP) of 10 000 µmol m$^{-2}$·s$^{-1}$ for 600 ms, then with a delay of 50 s, Actinic Light (AL) of 75 µmol m$^{-2}$·s$^{-1}$ was applied, followed by a set of SP. (g – **i**) Light curves of *Arabidopsis* pot cultures inoculated with *S. ruberrimus* with standard error. g – Y(II), h – NPQ, i – ETR. Actinic light of increasing irradiance (0, 11, 18, 27, 58, 100, 131, 221, 344, 536, 830 µmol m$^{-2}$·s$^{-1}$) was applied within 30 s phases. Saturating pulses lasted 600 ms.

However, UV-A impact on other leaves does not seem significant [84]. In a recent study, supplementation of white light with UV-A1 (in the range of 350–400 nm, similar to our conditions) results in growth limitations and flavonoid content depletion in soil-grown *Arabidopsis* [85]. We assessed the total fresh weight of *Arabidopsis* shoots and roots grown in *in vitro* conditions. The plants germinated in the presence of UV-A, which is also different from the previous studies [5,84]. In *in vitro* conditions of our study, the root length but not weight was inhibited by UV-A. Excess UV-A does not influence the dry root weight of the *Arabidopsis* Columbia ecotype (Col-4) [6].

Alterations in root growth and architecture are often the first visible manifestation of the interaction between the plant host and its microbiota. These growth modifications arise from: phytohormone and other metabolite synthesis by the microorganism and adaptation of plant hormonal homeostasis and developmental program to the symbiotic/endophytic partnership [86]. Here the only clear effect of co-culture of plants with endophytic microorganisms was altered root elongation and root width in the cases of *Arabidopsis* co-cultured with *P. chrysanthemicola, D. eres,* and *P. columnaris*. Additionally, *S. ruberrimus* had a similar effect on root thickness, but root length did not differ from the wild type. Numerous microorganisms produce IAA and interfere in plant auxin signalling, including the ones used in the study. Both *S. ruberrimus* and *Mucor* sp. have been previously shown to possess this ability [57,87]. Callose and lignin deposition in root cells of plants colonised by microorganisms has also been previously reported [88]. We aimed to determine whether UV-A radiation will affect these processes. We did not observe any effects of UV regarding this aspect of the interaction; the relationship between these parameters did not significantly differ between plants cultured in PAR and PAR supplemented with UV-A.

*P. chrysanthemicola* and *Mucor sp.* inhibited plant growth regardless of light conditions. This can be partially explained by the ambiguous behaviour of endophytic fungi. *P. chrysanthemicola,* has been described as a plant pathogen [89,90] and beneficial endophyte [91,92]. *Mucor* is a ubiquitous genus in many habitats. It is known as a saprotroph, colonizing dead plant material [93] and has been isolated as an endophyte on several occasions [46,94]. *Mucor* has been shown to improve plant growth [95] and cause mucormycosis in the human population [96]. We expected that UV-A supplementation, which may act as a stress factor for the plant, could cause the fungus to take advantage of the host and develop more rapidly within its tissues. We did not see such an effect, which indicates that UV-A could not promote pathogenesis. The other microorganisms used in this study were not able to protect the plant against the deleterious impact of UV-A radiation. In all examined cases, the fungus colonisation rates were similar in plants growing in visible light compared to plants cultured under light supplemented with UV-A, independent of the growth response. This shows that any effect of UV-A on the interaction between the host plant and its fungal symbiont is unlikely to stem from altered colonisation. The host plant-fungal symbiont interaction developed in high UV conditions [97] during the plant terrestrialization [98]. Arbuscular mycorrhizal and lichen symbiosis are believed to be pivotal for early plants in their water-to-land transitions by improving nutrient and water availability [99]. From the evolutionary perspective, the CRY-dependent blue/UV-A and UVR8-based UV-B photoreception has emerged in the oldest aquatic algae lines and has been well developed at the early stages of plant land conquer [100].

Our experiments performed on *Arabidopsis* grown in pot cultures showed that the flavonol content was affected by the interaction of UV-A radiation and the presence of endophytes. However, mutants defective in flavonol and sinapate content behaved similarly to wild-type plants, exhibiting growth retardation when inoculated with *P. chrysanthemicola* and *S.*

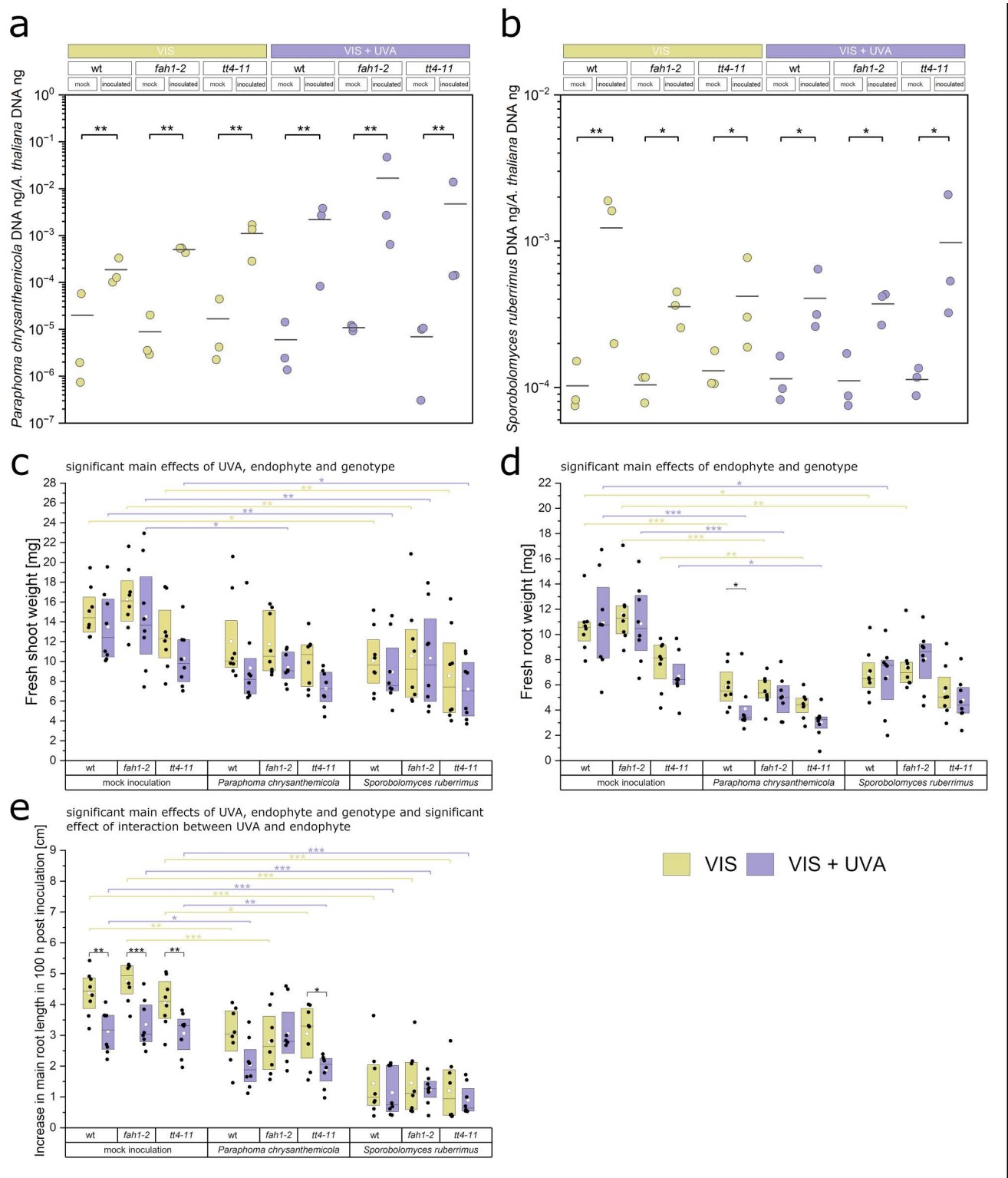

**Fig. 6. Endophyte inoculation of *Arabidopsis thaliana*, wild type, *fah1*, and *tt4* mutants, grown in the presence or absence of UV-A, assessed by the qPCR method. (a-b)** The ratio of fungal DNA to plant DNA in roots and shoots of plants inoculated with *Paraphoma chrysanthemicola* (a) and *Sporobolomyces ruberrimus* **(b)**. The experiment was performed in three biological replicates. Brackets show the significant differences in means

of log-transformed qPCR measurements between inoculated and mock-inoculated samples light conditions (* $0.01 < p \leq 0.05$, ** $0.001 < p \leq 0.01$, *** $p \leq 0.001$, adjusted for multiple comparisons with Hommel's method). The set of all tested contrasts is in the S2 Appendix, Table S6. **(c - d)** Fresh weight of shoots (c) and roots (d) 18-day-old plants, either mock-inoculated (control) or inoculated with *Paraphoma chrysanthemicola,* or *Sporobolomyces ruberrimus*. Weight was measured 128 h after inoculation **(e)** Main root length increment of *Arabidopsis thaliana,* wild type, *fah1,* and *tt4* mutants, 100 hours post inoculation with *Paraphoma chrysanthemicola,* or *Sporobolomyces ruberrimus*. Plants were grown in the presence or absence of UV-A. **(c – e)** The experiment was performed in 8 biological replicates (plates) for every combination of inoculation type and light conditions. Black brackets show the significant differences in means between plants inoculated with the same endophyte, but subject to different light conditions. Yellow (VIS) and violet (VIS + UV) brackets show significant differences in means between inoculated samples and the mock-inoculated control (* $0.01 < p \leq 0.05$, ** $0.001 < p \leq 0.01$, *** $p \leq 0.001$, adjusted for multiple comparisons with the Benjamin-Hochberg method). The set of all tested contrasts is in the S2 Appendix, Table S7.

*ruberrimus.* In an outdoor wavelength exclusion experiment, leaf epidermal UV absorbance, corresponding to the flavonoid content, has been enhanced in the presence of UV-A. An even stronger effect is observed when both UV-A and UV-B act simultaneously [101]. On the other hand, an indoor study [24] designed with a comparable to ours UV-A: PAR ratio demonstrates a larger impact of blue light in increasing the flavonoid content than that of UV-A radiation. Cryptochromes and UVR8 are responsible for eliciting the flavonoid content in response to UV-A radiation [24]. Phot1 plays a role in the promotion of cry-mediated anthocyanin accumulation for plants growing under both blue light and UV-A radiation, which increases the overall phenolic content [24]. Both cryptochromes cry1 and cry2 control the expression of the chalcone synthase gene (*CHS*), which encodes the key enzyme of the flavonoid biosynthesis pathway [102]. *CHS* expression is moderately regulated by UV-A in a cry1-dependent manner. When UV-A acts synergistically with UV-B radiation, *CHS* expression is upregulated very strongly in a cryptochrome-independent manner [78,103]. UV-A triggers the expression of *CHS* through UVR8 only in the absence of cryptochromes in the Landsberg *Arabidopsis* ecotype [21]. In our experiments, we observed only the small main effect of the endophyte on the *CHS* gene. Fungi are known to elevate anthocyanin levels in plants by activation of the jasmonic and salicylic pathways [104]. The expression of the *PAL1* gene was affected by the presence of the endophyte, but also by UV-A. A similar effect of UV-A is reported for the *PAL* gene from tomatoes [105]. In *Arabidopsis*, cry1 regulates the expression of *PAL* in blue light [106]. Thus, cryptochromes, especially cry1, are major players in activating flavonoid biosynthesis pathway genes.

We also considered the possibility that the interaction with microorganisms affects UV photoreceptor gene expression and thus may alter UV perception in plants. In several species important from the agronomic point of view, mutations that alter the expression levels of photoreceptor genes affect their physiological traits [107]. For instance, the downregulation of *CRY2* delayed flowering in *Oryza sativa* [108] and of CRY1a/b increased germination in *Hordeum vulgare* [109]. Increased gene expression of *CRY2* enhanced lycopene content and pigmentation of *Solanum lycopersicum* fruits [110], overexpression of *CRY1a* resulted in early flowering of *Glycine max* [111] and elevated expression of *CRY1* reduced plant height in *Brassica sp* [112].

Our results did not show any substantial differences in the expression levels of *UVR8* and both cryptochromes between plants supplemented and non-supplemented with UV-A. UV-A irradiation was previously shown to diminish the cry2 protein abundance at the protein level [113]. The *UVR8* gene expression was previously found to be UV-B regulated in *Colobanthus quitensis*, in which UV-B increases *UVR8* expression, but only in plants not inoculated with endophytes, suggesting their photoprotective role [83]. On the other hand, in Sauvignon blanc grape berries the *VvUVR8* gene is not significantly affected by UV-B radiation at any stage of development [114]. In *Arabidopsis*, changes in the UVR8 monomer-dimer equilibrium upon UV-B irradiation are of pivotal importance for photoreceptor signalling [115,116]. Much less is known about UVR8 activity, regulation, and expression under UV-A [22]. An outdoor experiment demonstrated that *UVR8* levels are unaffected by either UV-A or combined UV-A and UV-B treatments [101]. Phototropin gene expression is regulated by light [66], with *PHOT1* expression remaining unchanged under

short-term UV exposure [26], however, UV-A irradiation upregulates *PHOT2* transcript levels [117]. In our work, UV-A supplementation generally downregulated both *PHOT1* and *PHOT2* transcripts, and the presence of endophytes additionally influenced *PHOT2* levels. In the case of potato parasitic oomycete *Phytophthora infestans*, expression of Stphot1 and Stphot2 enhances the infection. While Stphot1 kinase activity is required for blue-light-mediated immune suppression, Stphot2 is not [118]. Thus, phototropins may play different roles in the susceptibility of plants to inoculation by microbes.

Our results suggest that the presence of UV-A in the light spectrum has a limited effect on the plant-endophyte interaction when the ratio of UV-A to PAR is close to the ratio in sunlight. This is essential due to the importance of plant growth-promoting microorganisms used in agriculture and the presence of UV-A in the natural environment. Thus, published experiments using standard growth chambers, without UV-A light, may provide physiologically relevant data. The supplementation with UV-A radiation in horticulture is still a matter of debate, as it has beneficial effects for some pollinators but enhances the sporulation of plant pathogens or reproduction of insects [29,32]. Plant endophytes may be used for biological pest control in greenhouses [119]. In such conditions, UV-A supplementation is considered less detrimental for microorganisms and may also exert its potential beneficial effects on plants [120]. The interactions between the microorganism and the host plant are very often species-specific. For instance, the host-plant interactions change even between different representatives of the cabbage family, which are very closely related to each other [56]. Our growth conditions were free from light and temperature fluctuations, which may influence the plant-endophyte interaction. Since very little is known about how UV affects the plant-microorganism interaction, we focused on this factor, acting in constant conditions. The use of *in vitro* cultures allowed us to reduce the risk of cross-contamination, but it led to the unphysiological irradiation of roots. Considering these experimental limitations and the complexity of signalling pathways, more research is needed not only on model plants but also on crop species and various endophyte taxa. From the local socio-economical point of view and previous experience, it would be worthwhile to continue studies on plant-endophyte interactions in edible *Brassicaceae* in the context of UV supplementation. Our research can be a good starting point for investigations aimed at including UV-A radiation in crop cultures. UV can be beneficial for plant growth by promoting the development of particular plant traits. UV, like other environmental factors, requires adaptation of plant metabolism which involves mechanisms related to plant microorganism cross-talk. In this study, we show that application of UV-A radiation in the range of 350–400 nm does not exert a substantial negative effect on the plant-microorganism interaction of the investigated species. This indicates that application of this additional factor in crop culture most probably will not affect the efficiency of plant growth promotion by microorganisms widely utilised in agriculture. It needs to be emphasized though, that the experimental setup used in this study exhibits drawbacks and requires verification in more natural conditions where plants are exposed to multiple factors that affect their response to both UV and endophytic microorganisms.

## Conclusions

In general, UV-A radiation, applied in this study in the 350–400 nm range, had no considerable impact on the level of plant colonization by the tested endophytes. Changes in light spectral properties did not substantially affect plant growth responses to endophytic fungi. In most cases, we found no evidence that the effect of inoculation with fungi on root biomass architecture of plants is modulated by UV-A treatment, as evidenced by the lack of significant interaction between the effects of endophyte inoculation and light conditions in most of the ANOVA results. However, UV-A acclimation influenced the effects of the presence of endophyte on the expression of the plant defence gene *PDF2.1*. Inoculation of plants with *Sporobolomyces ruberrimus* influenced the UV-A effect on the epidermal flavonols. Our study provides the first insight into the potential influence of UV-A on interactions between the model plant *Arabidopsis thaliana* and endophytes representing major taxonomic groups of fungi (Fig. 7).

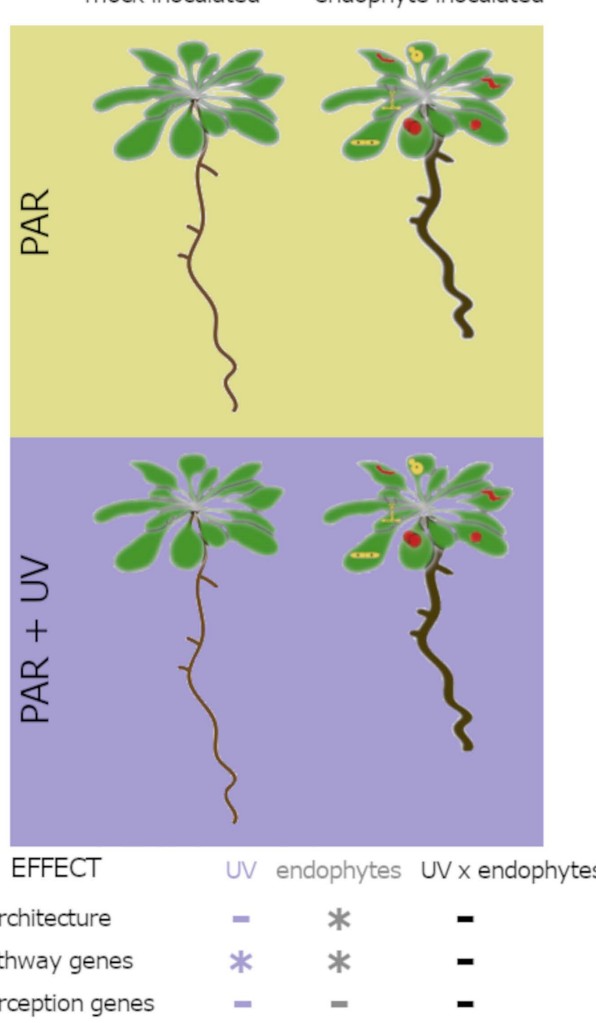

**Fig. 7. Supplementation of PAR with UV-A did not substantially affect the interaction between *Arabidopsis thaliana* and endophytic fungi.** Both UV-A and endophytes affected the expression of genes related to the salicylic acid pathways. The microorganisms did not affect plant UV-A perception. No endophyte-induced alteration in UV-A receptor expression and anthocyanin accumulation were observed. UV-A did not alter microorganism-dependent changes in root patterning. The minus represents no significant effect of UV-A or endophyte or interaction of both factors. The asterisk represents a significant effect as detected with ANOVA.

## Supporting information

**S1 Appendix. Primer sequences used for qPCR confirming endophyte presence and for real–time analysis of *Arabidopsis thaliana* gene expression.**
(DOCX)

**S2 Appendix. The results of the statistical analysis.** Tables S1-S7.
(DOCX)

**S1 Fig. Light spectra used in the experiment.** White illumination of ca. 225 µmol m$^{-2}$ s$^{-1}$ PAR was provided by LEDs (EPILEDs, 5000 K) supplemented with UV-A of 13 µmol m$^{-2}$ s$^{-1}$ from fluorescent tubes (Philips TL 6W BLB) covered with either Lee #226 gel filter (upper panel), or with a polyester film, Autostat CT5 (lower panel).
(TIFF)

**S2 Fig. *Arabidopsis thaliana* wild-type seedlings grown in the presence and absence of UV-A, non-inoculated or inoculated with *Paraphoma chrysanthemicola*, *Phomopsis columnaris*, *Diaporthe eres*, *Mucor* sp., and yeasts *Sporobolomyces ruberrimus* on the first day after inoculation (the 9$^{th}$ day of growth) (a).** (b - d) Length (b), average volume (c), and average diameter (d) of roots of non-inoculated plants on the 9$^{th}$ day of growth. The experiment consists of nine biological replicates (plates) per every combination of light conditions and inoculation type. In b, the bracket shows a significant difference in means (*** $p \leq 0.001$). The results of the statistical analysis are in the S2 Appendix, Table S5. White and black rectangles are 1 cm by 1 cm each.
(TIF)

**S3 Fig. *Arabidopsis thaliana* wild-type plants grown in the presence or absence of UV-A, non-inoculated or inoculated with *Paraphoma chrysanthemicola*, *Phomopsis columnaris*, *Diaporthe eres*, *Mucor* sp., and yeasts *Sporobolomyces ruberrimus* ten days after inoculation (18$^{th}$ day of growth).** White and black rectangles are 1 cm by 1 cm each.
(TIF)

**S4 Fig. Shoots of *Arabidopsis* inoculated with endophytes *Paraphoma chrysanthemicola*, *Phomopsis columnaris*, *Diaporthe eres*, *Mucor* sp. and *Sporobolomyces ruberrimus* in the presence or absence of UV-A radiation.** Endophytes were stained Wheat Germ Agglutinin conjugated with Texas Red. Images, recorded with a laser scanning confocal microscope, show transmitted light channel merged with a red fluorescence channel. Plants not inoculated with the endophyte serve as a control for plant tissue autofluorescence. Scale bars = 50 µm.
(TIF)

**S5 Fig. Relative transcript levels of *CRY1* (a), *CRY2* (b), *PHOT1* (c), *PHOT2* (d), and *UVR8* (e) genes in leaves of plants inoculated with *Paraphoma chrysanthemicola*, *Phomopsis columnaris*, *Diaporthe eres*, *Mucor* sp., *Sporobolomyces ruberrimus* grown in the presence or absence of UV-A.** The experiment was performed in nine biological replicates for *PHOT1*, and *PHOT2* and six biological replicates for *CRY1, CRY2*, and *UVR8*. The results of the statistical analysis can be found in the S2 Appendix, Table S3.
(TIF)

**S6 Fig. Increment of the main root length of *Arabidopsis thaliana,* wild type, *fah1–2,* and *tt4–11* mutants, assessed every 12 hours post inoculation with *Paraphoma chrysanthemicola,* or *Sporobolomyces ruberrimus*.** Plants were grown in the presence or absence of UV-A. The experiment was performed in 8 biological replicates (independent plates) for every combination of inoculation type and light conditions.
(TIF)

## Acknowledgments

We would like to thank Anna Kozłowska-Mroczek for her excellent assistance in preparing the plant material. We thank Prof. Pedro Aphalo (University of Helsinki) for his helpful comments on the manuscript.

## Author contributions

**Conceptualization:** Rafał Ważny, Agnieszka Domka, Piotr Rozpądek, Justyna Łabuz.

**Data curation:** Justyna Łabuz.

**Formal analysis:** Paweł Hermanowicz.

**Funding acquisition:** Justyna Łabuz.

**Investigation:** Aleksandra Giza, Paweł Hermanowicz.

**Supervision:** Justyna Łabuz.

**Writing – original draft:** Piotr Rozpądek, Justyna Łabuz.

**Writing – review & editing:** Piotr Rozpądek, Justyna Łabuz.

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
