## [Decision Letter · Decision Letter 0]

16 Feb 2025

PONE-D-24-56475Effect of UV-A on Endophyte Colonisation of Arabidopsis thalianaPLOS ONE

Dear Dr. Łabuz,

Thank you for submitting your manuscript to PLOS ONE. After careful consideration, we feel that it has merit but does not fully meet PLOS ONE’s publication criteria as it currently stands. Therefore, we invite you to submit a revised version of the manuscript that addresses the points raised during the review process.

We look forward to receiving your revised manuscript.

Kind regards,

Eugenio Llorens

Academic Editor

PLOS ONE

Journal Requirements:

“This research was funded by the Priority Research Area BioS under the program Excellence Initiative – Research University at the Jagiellonian University in Krakow, project "The role of ultraviolet radiation in the colonisation of plants by endophytes", number B.1.11.2020.74.”

Reviewers' comments:

Reviewer's Responses to Questions

**Comments to the Author**

1. Is the manuscript technically sound, and do the data support the conclusions?

Reviewer #1: Yes

Reviewer #2: Partly

Reviewer #3: Yes

2. Has the statistical analysis been performed appropriately and rigorously? 

Reviewer #1: Yes

Reviewer #2: Yes

Reviewer #3: Yes

3. Have the authors made all data underlying the findings in their manuscript fully available?

Reviewer #1: Yes

Reviewer #2: Yes

Reviewer #3: Yes

4. Is the manuscript presented in an intelligible fashion and written in standard English?

Reviewer #1: Yes

Reviewer #2: Yes

Reviewer #3: Yes

5. Review Comments to the Author

Reviewer #1: The authors investigate the impact of UV-A on endophyte colonisation of Arabidopsis thaliana of fungi from different taxonomic groups. They found that physiologically relevant levels of UV-A did not significantly affect the conlonisation of shoots and roots by endophytic fungi, while affect the expression of genes associated with symbiosis establishment process. I have three major suggestions for this study. 1. The authors should provide a phylogenetic tree to show the position of each endophytic fungus. 2. the authors should provide some information about the ecological significance of UV and plant-microbe symbiosis from the plant terrestrialization. The UV is an important factor during plant terrestrialization. 3. I think the conclusion section should be re-written. The conclusion section should include the main findings and significance of the study. Overall, this manuscript is complete and interesting.

Reviewer #2: This study investigates the effects of UV-A radiation on the interaction between Arabidopsis thaliana and endophytic fungi. The overall experimental design is well-structured and the data is rich; however, there are several areas that could be improved. Below are specific questions and suggestions for revision, concerning the experimental design, data analysis, discussion, and other aspects, aiming to enhance the quality of the research and the paper:

1. Although the proportion of UV-A radiation to Photosynthetically Active Radiation (PAR) was set to closely resemble natural sunlight, other environmental conditions in the growth chamber may still differ significantly from natural environments. For example, factors such as temperature fluctuations, humidity variations, and the combined effect of other environmental factors were not adequately simulated in the experiment. These differences could affect the generalizability of the results. It is recommended that the authors discuss how these discrepancies may influence the plant-endophyte interaction or supplement the study with experiments conducted under more natural environmental conditions to enhance the applicability of the conclusions.

2. There are differences in the inoculation methods and concentrations used for different endophytes. For instance, Paraphoma chrysanthemicola, Phomopsis columnaris, and Diaporthe eres were inoculated using mycelial blocks, while Mucor sp. was inoculated with spores, and Sporobolomyces ruberrimus was inoculated with yeast cell suspensions. Additionally, the specific inoculation quantities are described inconsistently. These variations may interfere with the accurate comparison of the effects of UV-A on different endophytes. It is suggested that the authors standardize the inoculation concentration reporting methods, or conduct additional experiments to investigate the influence of different inoculation concentrations on the results, in order to confirm the reliability of the current inoculation methods and concentrations.

3. In the discussion section, some interpretations of the results are inconsistent with previous studies. For example, in interpreting the effects of UV-A on plant growth, there are discrepancies between the current findings and those in other studies regarding the promotion or inhibition of plant growth by UV-A. However, the authors did not thoroughly explore the reasons for these differences.

4. Based on the experimental results, the authors may consider constructing a conceptual model to integrate the interactions between UV-A, endophytes, and plants. This would help to clarify the relationships between the different factors and how they collectively influence plant growth, defense mechanisms, and metabolic processes.

5. Given that the study involves plant-microbe interactions in agriculture and the impact of UV-A radiation, it is recommended that the authors further explore the practical applications of their findings in agricultural production. For example, how you results might be used to optimize greenhouse cultivation conditions or how an understanding of the relationship between UV-A and endophytes could contribute to the development of new agricultural strategies to improve crop yield and quality, thereby increasing the practical relevance of the study.

6. The UV-A radiation intensity used in the study was 13 µmol m⁻² s⁻¹, but the rationale for selecting this specific intensity was not explained. The authors are encouraged to provide a justification for this choice, indicating whether it corresponds to natural UV-A intensity levels or if there is supporting literature that highlights the physiological relevance of this particular intensity.

7. While the paper measures parameters such as fresh weight, root length, and root volume, there is limited discussion of how changes in these parameters may be related to endophyte colonization or UV-A radiation. It is recommended that the authors conduct a more in-depth analysis of whether these parameter changes are directly associated with endophytic colonization or UV-A exposure and provide more physiological explanations for these observations.

Reviewer #3: Authors presented interesting point of view and concentrated on UV-A influenced on Arabidopsis fungal and yeast endophytes. Authors stated that UV-A upregulated the expression of genes involved in the establishment of symbiosis. Authors confirmed that there were no significant correlation between UV-A and the presence of endophytes on other examined plant traits, including plant fresh weight, root system architecture, and expression of plant photoreceptor genes. Moreover, UV-A does not directly influence plant colonisation by endophytes. Furthermore, it does trigger the upregulation of plant defence genes during interaction with selected fungal and yeast.

Introduction gives the reader sufficient background to analyse Author’s obtained results.

Materials and methods are described in details, in a repetitive way. Please, add deeply explanation (maybe in M&M section) why Authors decide exactly used fah1-2 (ferulic acid 5-hydroxylase 1) and tt4-11 (chalcone synthase) mutants and what are their characteristics – only on page 24 in the manuscript we can find that these genes can be induced by UV radiation.

Did using sodium hypochlorite solution with Tween before collecting samples in liquid nitrogen not produce some artefacts in qPCR endophytes quantification?

Please, work a bit on resolution (quality) of photographic documentation in the whole Figure 1;

Unfortunately, after downloading at 100% enlargements almost all markings on the figures have too low resolution and it was difficult to read.

On the other side, Authors in almost all aspects used scatterplots, which in my humble opinion are very difficult in analyses of real statistical significance.

Taking into account that Authors presented mainly “negative” correlations between many/main factors in experiments discussion part of the manuscript is good rethink and well-conducted.

Moreover, Authors concluded, that UV-A in the light spectrum has a limited effect on the plant-endophyte interaction in their experiments, therefore, I suggest to add some future prospects coming from that results as well as defining new research paths.

6. PLOS authors have the option to publish the peer review history of their article (what does this mean? ). If published, this will include your full peer review and any attached files.

**Do you want your identity to be public for this peer review?** For information about this choice, including consent withdrawal, please see our Privacy Policy .

Reviewer #1: No

Reviewer #2: No

Reviewer #3: No

---

## [Author Response · Author response to Decision Letter 1]

28 Mar 2025

Thank you very much for the reviews. We are providing a point-to-point response.

Review Comments to the Author

Reviewer #1: The authors investigate the impact of UV-A on endophyte colonisation of Arabidopsis thaliana of fungi from different taxonomic groups. They found that physiologically relevant levels of UV-A did not significantly affect the conlonisation of shoots and roots by endophytic fungi, while affect the expression of genes associated with symbiosis establishment process. I have three major suggestions for this study.

Thank you very much for all suggestions.

1. The authors should provide a phylogenetic tree to show the position of each endophytic fungus.

We have prepared a phylogenetic tree of the fungi and incorporated it as a new part of Fig.1. The following description has been added to the Materials and Methods section:

“The phylogeny was inferred using the Maximum Likelihood method and Tamura-Nei (1993) model of nucleotide substitutions, and the tree with the highest log likelihood (-20 328.52) is shown. The initial tree for the heuristic search was selected by choosing the tree with the superior log-likelihood between a Neighbor-Joining (NJ) tree and a Maximum Parsimony (MP) tree. The NJ tree was generated using a matrix of pairwise distances computed using the p-distance. The MP tree had the shortest length among 10 MP tree searches, each performed with a randomly generated starting tree. The analytical procedure encompassed 25 coding nucleotide sequences using 1st, 2nd, 3rd, and non-coding positions with 2577 positions in the final dataset. Evolutionary analyses were conducted in MEGA12 utilizing up to 8 parallel computing threads.”

2. the authors should provide some information about the ecological significance of UV and plant-microbe symbiosis from the plant terrestrialization. The UV is an important factor during plant terrestrialization.

The following paragraph was added to the discussion section:

“The host plant-fungal symbiont interaction developed in high UV conditions (86) during the plant terrestrialization (87). Arbuscular mycorrhizal and lichen symbiosis are believed to be pivotal for early plants in their water-to-land transitions by improving nutrient and water availability (88). From the evolutionary perspective, the CRY-dependent blue/UV-A and UVR8-based UV-B photoreception has emerged in the oldest aquatic algae lines and has been well developed at the early stages of plant land conquer (89).”

3. I think the conclusion section should be re-written. The conclusion section should include the main findings and significance of the study.

We have rewritten the conclusions, and now it reads:

“In general, UV-A radiation had no considerable impact on the level of plant colonization by the tested endophytes. Changes in light spectral properties did not affect plant growth responses to endophytic fungi. In most cases, we found no evidence that the effect of inoculation with fungi on root biomass architecture of plants is modulated by UV-A treatment, as evidenced by the lack of significant interaction between the effects of endophyte inoculation and light conditions in most of the ANOVA results. However, UV-A acclimation influenced the effects of the presence of endophyte on the expression of the plant defence gene PDF2.1. Inoculation of plants with Sporobolomyces ruberrimus influenced the UV-A effect on the epidermal flavonols. Our study provides the first insight into the potential influence of UV-A on interactions between the model plant Arabidopsis thaliana and endophytes representing major taxonomic groups of fungi (Fig. 7).”

Overall, this manuscript is complete and interesting.

Thank you very much for this comment.

Reviewer #2: This study investigates the effects of UV-A radiation on the interaction between Arabidopsis thaliana and endophytic fungi. The overall experimental design is well-structured and the data is rich; however, there are several areas that could be improved. Below are specific questions and suggestions for revision, concerning the experimental design, data analysis, discussion, and other aspects, aiming to enhance the quality of the research and the paper:

Thank you very much for the remarks on our manuscript. We hope that it improved after our revision.

1. Although the proportion of UV-A radiation to Photosynthetically Active Radiation (PAR) was set to closely resemble natural sunlight, other environmental conditions in the growth chamber may still differ significantly from natural environments. For example, factors such as temperature fluctuations, humidity variations, and the combined effect of other environmental factors were not adequately simulated in the experiment. These differences could affect the generalizability of the results. It is recommended that the authors discuss how these discrepancies may influence the plant-endophyte interaction or supplement the study with experiments conducted under more natural environmental conditions to enhance the applicability of the conclusions.

We have added the following paragraph to address the limitation of our study:

“Our growth conditions were free from light and temperature fluctuations, which may influence the plant-endophyte interaction. Since very little is known about how UV affects the plant-microorganism interaction, we focused on this factor, acting in constant conditions. The use of in vitro cultures allowed us to reduce the risk of cross-contamination, but it led to the unphysiological irradiation of roots. Considering these experimental limitations and the complexity of signalling pathways, more research is needed not only on model plants but also on crop species and various endophyte taxa.”

In addition, we have made it clearer in the manuscript that UV-A applied in our study was in the UV-A1 (350 – 400 nm) waveband. Thus, we cannot exclude the possibility that the application of UV-A in the A2 range (315-350) may exert more pronounced effects. UV-A2 can be perceived to some extent by the UVR-8 photoreceptor, so its effects may be different and resemble more the effects of UV-B.

2. There are differences in the inoculation methods and concentrations used for different endophytes. For instance, Paraphoma chrysanthemicola, Phomopsis columnaris, and Diaporthe eres were inoculated using mycelial blocks, while Mucor sp. was inoculated with spores, and Sporobolomyces ruberrimus was inoculated with yeast cell suspensions. Additionally, the specific inoculation quantities are described inconsistently. These variations may interfere with the accurate comparison of the effects of UV-A on different endophytes. It is suggested that the authors standardize the inoculation concentration reporting methods, or conduct additional experiments to investigate the influence of different inoculation concentrations on the results, in order to confirm the reliability of the current inoculation methods and concentrations.

The microorganisms used in this study exhibit diverse lifestyles. Paraphoma chrysanthemicola, Phomopsis columnaris, and Diaporthe eres are slow-growing species. Mucor is a fast-growing and abundantly sporulating species, while Sporobolomyces is a yeast with slow growth on a solid medium. We agree that there is an inconsistency in the description of inoculation methods, however, the most reliable method of inoculation depends on the properties of individual species. The selected inoculation methods led to successful colonisation, as shown by the qPCR results

3. In the discussion section, some interpretations of the results are inconsistent with previous studies. For example, in interpreting the effects of UV-A on plant growth, there are discrepancies between the current findings and those in other studies regarding the promotion or inhibition of plant growth by UV-A. However, the authors did not thoroughly explore the reasons for these differences.

We added the following paragraph to the discussion:

“Our data showed that UV-A reduced the fresh weight of Arabidopsis shoots regardless of endophyte presence. UV-A has a positive impact on the rosette diameter of several Arabidopsis ecotypes grown in soil (5). It also elevates the area and blade length of Arabidopsis Columbia mature leaves grown in soil culture. However, UV-A impact on other leaves seems not significant (77). In a recent study, supplementation of white light with UV-A1 (in the range of 350–400 nm, similar to our conditions) results in growth limitations and flavonoid content depletion in soil-grown Arabidopsis (78). We assessed the total fresh weight of Arabidopsis shoots and roots grown in in vitro conditions. The plants germinated in the presence of UV-A, which is also different from the previous studies (5), (77). In in vitro conditions of our study, the root length but not weight was inhibited by UV-A. Excess UV-A does not influence dry root weight of the Arabidopsis Columbia ecotype (6).”

4. Based on the experimental results, the authors may consider constructing a conceptual model to integrate the interactions between UV-A, endophytes, and plants. This would help to clarify the relationships between the different factors and how they collectively influence plant growth, defense mechanisms, and metabolic processes.

We have proposed a model in Fig. 7, described in the Conclusions section.

5. Given that the study involves plant-microbe interactions in agriculture and the impact of UV-A radiation, it is recommended that the authors further explore the practical applications of their findings in agricultural production. For example, how you results might be used to optimize greenhouse cultivation conditions or how an understanding of the relationship between UV-A and endophytes could contribute to the development of new agricultural strategies to improve crop yield and quality, thereby increasing the practical relevance of the study.

We have added additional information to the final paragraph of this topic:

“Plant endophytes may be used for biological pest control in greenhouses (109). In such conditions, UV-A supplementation is considered less detrimental for microorganisms and may also exert its potential beneficial effects on plants (110).”

6. The UV-A radiation intensity used in the study was 13 µmol m⁻² s⁻¹, but the rationale for selecting this specific intensity was not explained. The authors are encouraged to provide a justification for this choice, indicating whether it corresponds to natural UV-A intensity levels or if there is supporting literature that highlights the physiological relevance of this particular intensity.

Our UV-A: PAR photon ratio was ca. 0.052. The exact ratio of the UV-A waveband used in the study (350 – 400 nm) to total PAR (400 – 700 nm) in sunlight depends on the atmospheric conditions. The standard 1.5 atmospheric thickness sunlight spectrum gives a photon ratio of 0.0515 (American Society for Testing and Materials, 2012. ASTM G173 - 03 Standard Tables for Reference Solar Spectral Irradiances: Direct Normal and Hemispherical on 37° Tilted Surface). This spectrum is a good representation of the sunlight spectrum when the sky is clear, at mid-latitudes during the vegetation period. We have added this information to the manuscript.

Similar irradiation schemes were used in the following papers, cited in the manuscript:

Sun X, Kaiser E, Aphalo PJ, Marcelis LFM, Li T. 2024. Plant responses to UV-A1 radiation are genotype and background irradiance-dependent. Environmental and Experimental Botany 219, 105621.

Zhang Y, Sun X, Aphalo PJ, Zhang Y, Cheng R, Li T. 2024. Ultraviolet‐A1 radiation induced a more favorable light‐intercepting leaf‐area display than blue light and promoted plant growth. Plant, Cell and Environment 47, 197–212.

7. While the paper measures parameters such as fresh weight, root length, and root volume, there is limited discussion of how changes in these parameters may be related to endophyte colonization or UV-A radiation. It is recommended that the authors conduct a more in-depth analysis of whether these parameter changes are directly associated with endophytic colonization or UV-A exposure and provide more physiological explanations for these observations.

We addressed the comment by the following paragraph:

“Alterations in root growth and architecture are often the first visible manifestation of the interaction between the plant host and its microbiota. These growth modifications arise from: phytohormone and other metabolite synthesis by the microorganism and adaptation of plant hormonal homeostasis and developmental program to the symbiotic/endophytic partnership. Here the only clear effect of co-culture of plants with endophytic microorganisms was altered root elongation and root width in the cases of Arabidopsis co-cultured with P. chrysanthemicola, D. eres, and P. columnaris. Additionally, S. ruberrimus had a similar effect on root thickness, but root length did not differ from the wild type. Numerous microorganisms produce IAA and interfere in plant auxin signalling, including the ones used in the study. Both S. ruberrimus and Mucor sp. have been previously shown to possess this ability. Callose and lignin deposition in root cells of plants colonised by microorganisms has also been previously reported. We aimed to determine whether UV-A radiation will affect these processes. We did not observe any effects of UV regarding this aspect of the interaction; the relationship between these parameters did significantly differ between plants cultured in PAR and PAR supplemented with UV-A.”

Reviewer #3: Authors presented interesting point of view and concentrated on UV-A influenced on Arabidopsis fungal and yeast endophytes. Authors stated that UV-A upregulated the expression of genes involved in the establishment of symbiosis. Authors confirmed that there were no significant correlation between UV-A and the presence of endophytes on other examined plant traits, including plant fresh weight, root system architecture, and expression of plant photoreceptor genes. Moreover, UV-A does not directly influence plant colonisation by endophytes. Furthermore, it does trigger the upregulation of plant defence genes during interaction with selected fungal and yeast.

Introduction gives the reader sufficient background to analyse Author’s obtained results. Materials and methods are described in details, in a repetitive way.

Thank you very much for all your comments on the manuscript.

Please, add deeply explanation (maybe in M&M section) why Authors decide exactly used fah1-2 (ferulic acid 5-hydroxylase 1) and tt4-11 (chalcone synthase) mutants and what are their characteristics – only on page 24 in the manuscript we can find that these genes can be induced by UV radiation.

We added the following explanation to the M&M section:

“The tt4-11 mutant is deficient in the CHS protein, thus lacking flavonoids, with increased sinapic acid ester content (48). The fah1-2 mutant lacks the F5H, ferulate-5-hydroxylaseis, and is deficient in sinapoylmalate and sinapoyl choline (49), required for plant defence (33) and UV-B protection (50) (46).”

Did using sodium hypochlorite solution with Tween before collecting samples in liquid nitrogen not produce some artefacts in qPCR endophytes quantification?

The sodium hypochlorite solution with Tween was used to remove any loosely bound bacteria or fungi from the surface of the plants so that only those residing inside the tissues (endophytes) were quantified by qPCR. The procedure was optimized for Arabidopsis in vitro cultures to limit (as much as possible) tissue degradation.

Please, work a bit on resolution (quality) of photographic documentation in the whole Figure 1;

Unfortunately, after downloading at 100% enlargements almost all markings on the figures have too low resolution and it was difficult to read.

We have prepared a new Fig.1, which now also includes the phylogenetic tree requested by Reviewer 1. We are sorry that the resolution of the figures was too low. However, it may be due to the PloSOne submission system. The figures can be downloaded separately using the link above each figure.

On the other side, Authors in almost all aspects used scatterplots, which in my humble opinion are very difficult in analyses of real statistical significance.

We have decided to use box

---

## [Decision Letter · Decision Letter 1]

11 Apr 2025

Effect of UV-A on Endophyte Colonisation of Arabidopsis thaliana

PONE-D-24-56475R1

Dear Dr. Łabuz,

We’re pleased to inform you that your manuscript has been judged scientifically suitable for publication and will be formally accepted for publication once it meets all outstanding technical requirements.

Kind regards,

Eugenio Llorens

Academic Editor

PLOS ONE

Reviewers' comments:

Reviewer's Responses to Questions

**Comments to the Author**

1. If the authors have adequately addressed your comments raised in a previous round of review and you feel that this manuscript is now acceptable for publication, you may indicate that here to bypass the “Comments to the Author” section, enter your conflict of interest statement in the “Confidential to Editor” section, and submit your "Accept" recommendation.

Reviewer #1: All comments have been addressed

Reviewer #2: All comments have been addressed

Reviewer #3: All comments have been addressed

2. Is the manuscript technically sound, and do the data support the conclusions?

Reviewer #1: Yes

Reviewer #2: Yes

Reviewer #3: Yes

3. Has the statistical analysis been performed appropriately and rigorously? 

Reviewer #1: (No Response)

Reviewer #2: Yes

Reviewer #3: Yes

4. Have the authors made all data underlying the findings in their manuscript fully available?

Reviewer #1: Yes

Reviewer #2: Yes

Reviewer #3: Yes

5. Is the manuscript presented in an intelligible fashion and written in standard English?

Reviewer #1: Yes

Reviewer #2: Yes

Reviewer #3: Yes

6. Review Comments to the Author

Reviewer #1: The authors have addressed my questions and comments well. I have only one suggestion. I think the title could be changed to "Effects of UV-A on Fungal Endophyte Colonisation of Arabidopsis thaliana". It helps readers focus on fungal endophyte rather than bacterial endophytes.

Reviewer #2: (No Response)

Reviewer #3: Authors significantly improved manuscript as well as response to all question. Therefore, I decide to accpet manuscript in current form.

7. PLOS authors have the option to publish the peer review history of their article (what does this mean? ). If published, this will include your full peer review and any attached files.

**Do you want your identity to be public for this peer review?** For information about this choice, including consent withdrawal, please see our Privacy Policy .

Reviewer #1: No

Reviewer #2: No

Reviewer #3: No

---

## [Editor Report · Acceptance letter]

PONE-D-24-56475R1

PLOS ONE

Dear Dr. Łabuz,

I'm pleased to inform you that your manuscript has been deemed suitable for publication in PLOS ONE. Congratulations! Your manuscript is now being handed over to our production team.

Kind regards,

on behalf of

Dr. Eugenio Llorens

Academic Editor

PLOS ONE